# GIM: IMPROVED INTERPRETABILITY FOR LARGE LANGUAGE MODELS

## ABSTRACT

Ensuring faithful interpretability in large language models is imperative for trustworthy and reliable AI. A key obstacle is self-repair, a phenomenon where networks compensate for reduced signal in one component by amplifying others, masking the true importance of the ablated component. While prior work attributes self-repair to layer normalization and back-up components that compensate for ablated components, we identify a novel form of self-repair that occurs within the attention mechanism, where softmax redistribution conceals the influence of important attention scores. This attention self-repair leads traditional ablation and gradient-based methods to underestimate the significance of all components contributing to these attention scores. We introduce Gradient Interaction Modifications (GIM), a technique that accounts for self-repair during backpropagation. Extensive experiments across multiple large language models (Gemma 2B/9B, LLAMA 1B/3B/8B, Qwen 1.5B/3B) and diverse tasks demonstrate that GIM frequently achieves state-of-the-art results on circuit identification and feature attribution. Our work is a significant step toward better understanding the inner mechanisms of LLMs, which is crucial for improving them and ensuring their safety. Our code is available at https://anonymous.4open.science/r/explainable_transformer-D693/README.md.

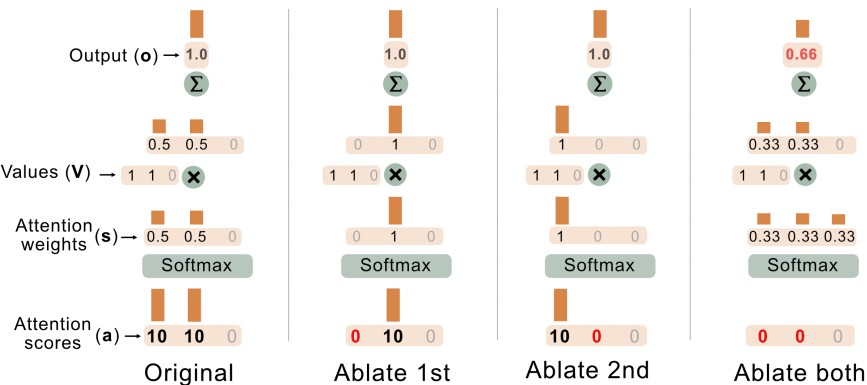

Figure 1: Attention self-repair. When multiple values associated with large attention weights contain similar information, ablating one attention score has little effect on the output because the softmax activation function compensates by increasing the weight of other positions. This results in perturbation-based and gradient-based explanation methods underestimating the importance of components that contribute to the attention scores through the keys and queries.

## 1 INTRODUCTION

Large language models (LLMs) have demonstrated remarkable capabilities across diverse tasks, from text summarization to code synthesis Brown et al. (2020). Despite their impressive performance, our understanding of their inner mechanisms remains limited. This opacity hinders our ability to

explain model outputs, catch hallucinations, systematically improve performance, and ensure reliable behavior in novel situations. To address this, most of the previous explainability methods aim to quantify how internal components (e.g., attention heads, neurons, token embeddings) affect model output Li et al. (2023). Two of the most widely adopted approaches, perturbation-based and gradient-based methods, both rest on the assumption that modifying a truly salient component will lead to a commensurate change in the model's output. Perturbation-based methods (e.g., SHAP, LIME, activation patching) investigate the effect of large perturbations Lundberg & Lee (2017); Meng et al., while gradient-based methods analyze infinitesimally small perturbations Sundararajan et al. (2017). However, this core assumption that perturbing important components will perturb the model output often fails due to what is known as the *self-repair effect* Rushing & Nanda (2024).

Self-repair is a phenomenon in language models where downstream model components compensate for perturbed components, resulting in deceptively small output changes Wang et al. (2022); McGrath et al. (2023); Rushing & Nanda (2024). As a consequence, explanation methods may underestimate the importance of certain model components Rushing & Nanda (2024).

In this paper, we identify a new form of self-repair within the attention mechanism of LLMs. When multiple value vectors associated with large attention weights contain similar information, perturbing any individual attention score has little effect on the model output. If one position's attention score is reduced, the softmax activation function increases the attention weights of the other positions, maintaining similar output signals (see Figure 1). This attention self-repair systematically causes both perturbation-based methods and gradient methods to underestimate the importance of all components contributing to these attention scores.

To address these challenges, we propose **gradient interaction modifications** (GIM), a novel gradient-based attribution method specifically designed to account for self-repair effects in language models. GIM introduces targeted modifications to backpropagation to better handle interactions that cause self-repair. First, we develop **temperature-adjusted softmax gradients** to address the attention self-repair problem. Second, we implement **layernorm freeze** to deal with the self-repair in layer normalization Ali et al. (2022); Rushing & Nanda (2024). Finally, we incorporate **gradient normalization** as proposed by Achtibat et al. (2024), which we found essential for making our backpropagation modifications effective in practice.

Our key contributions are:

1. We identify and formalize the **attention self-repair** problem in softmax operations, showing how it fundamentally undermines the faithfulness metrics of traditional interpretability methods.

2. We introduce **GIM**, a novel gradient-based attribution method that combines three complementary techniques to address feature interactions in transformer components. We demonstrate that GIM is often more faithful than other gradient-based circuit identification and feature attribution methods across multiple large language models (Gemma 2B/9B, LLAMA 1B/3B/8B, and Qwen 1.5B/3B) and six datasets spanning four tasks (question-answering, fact-verification, sentiment classification, and hate speech detection).

3. We empirically isolate and analyze the importance of each of our three modifications across multiple large language models and tasks.

## 2 RELATED WORK

### 2.1 SELF-REPAIR IN NEURAL NETWORKS

Self-repair occurs when ablating components has unexpectedly small effects on model outputs because other components compensate for the ablation. Several studies have identified attention heads that suppress the signal from earlier layers, and the suppression is reduced when those early layers are ablated McGrath et al. (2023); Wang et al. (2022); McDougall et al. (2023). McGrath et al. (2023) identified what they called MLP erasure neurons that have a similar suppressing effect on the signal from previous layers. Rushing & Nanda (2024) investigated the role of layer normalization in self-repair. They demonstrated that normalization can compensate for ablated layers by rescaling the remaining signals. Our work identifies a new form of self-repair that occurs within attention

mechanisms, where softmax redistribution conceals the influence of important attention scores. We also show that this self-repair mechanism can impact the gradient.

## 2.2 GRADIENT-BASED ATTRIBUTION METHODS

Attribution methods quantify model components' contribution to specific model behaviors. Feature attribution methods quantify the importance of input features (e.g., tokens), while circuit identification methods quantify the importance of internal model components (e.g., attention heads). Most attribution methods are perturbation-based or gradient-based Lyu et al. (2024). Perturbation-based approaches (e.g., SHAP Lundberg & Lee (2017) and activation patching Meng et al. (2022)) attribute importance by measuring output changes when perturbing components. However, they require numerous forward passes, making them prohibitively slow for large language models. Gradient-based methods offer a faster alternative by leveraging backpropagation Sundararajan et al. (2017); Simonyan et al. (2014). However, they often produce unfaithful explanations due to neural network non-linearities, where first-order approximations fail to capture true component importance Achtibat et al. (2024); Kramár et al. (2024a). Methods such as Layer-wise Relevance Propagation (LRP) Bach et al. (2015), Integrated Gradients Sundararajan et al. (2017), and DeepLIFT Shrikumar et al. (2017b) attempt to address these limitations by modifying gradient flows, but empirically struggle with language models Edin et al. (2025; 2024); Achtibat et al. (2024).

Several studies propose specialized backpropagation rules for the transformer architecture. TransformerLRP Ali et al. (2022) treats the layer normalization scaling variable and the attention matrix as constants, essentially not backpropagating through them. ATP* Kramár et al. (2024b) propose a backpropagation rule for the softmax operation that better estimates the impact of large attention score perturbations. However, ATP* ignores the normalization contribution in the softmax operation. Achtibat et al. (2024) propose AttnLRP, which is a collection of backpropagation rules. Grad norm, their most important contribution to our work, divides the gradient by $N$ when backpropagating through operations where $N$ variables are multiplied together.[1] Using Taylor Decomposition and Shapley, they provide theoretical justification for this rule. While TransformerLRP and AttnLRP show that the combination of their backpropagation rules improves the attribution faithfulness, they do not isolate each rule's contribution to the improvements.

Unlike previous work, our approach specifically targets self-repair in attention mechanisms and systematically evaluates both existing and novel backpropagation modifications, measuring their individual and combined impacts on attribution faithfulness.

## 3 ATTENTION SELF-REPAIR

Self-repair in the attention layer is when an attention score strongly influences the output, yet perturbing it has minimal effect. This effect deems both perturbation- and gradient-based attribution methods inaccurate. In this section, we show when and why attention self-repair occurs and its consequences on gradient-based explanation methods.

In transformer models, attention mechanisms control information flow between positions. Position $i$ gathers information from all positions using the following computations:

$$
\begin{aligned}
\boldsymbol{a} &= \boldsymbol{Q}_i \cdot \boldsymbol{K}^T \quad \text{(Attention scores for position } i) \\
\boldsymbol{s} &= \text{Softmax}(\boldsymbol{a}) \quad \text{(Attention weights for position } i) \\
\boldsymbol{o} &= \boldsymbol{s} \cdot \boldsymbol{V} \quad \text{(Attention output for position } i)
\end{aligned}
\tag{1}
$$

where $\boldsymbol{Q}_i$ is the query for position $i$, $\boldsymbol{K}$ are the keys, and $\boldsymbol{V}$ are the values. The softmax function normalizes attention scores into a probability distribution:

$$
\text{Softmax}(\boldsymbol{a}) = \frac{e^{\boldsymbol{a}/\tau}}{\sum_k e^{\boldsymbol{a}_k/\tau}}
\tag{2}
$$

where $\tau$ is the temperature parameter. For clarity, we distinguish between softmax input "attention scores" $\boldsymbol{a}$ and softmax output "attention weights" $\boldsymbol{s}$.

---

[1] Achtibat et al. refer to grad norm as *the uniform rule*

Attention mechanisms can be viewed as information routing systems, where each value vector $V_j$ contains information from position $j$, and the attention weights $s_j$ determine how much information to copy from that position.

Self-repair occurs when multiple values with large attention weights contain similar information. When an attention score $a_j$ is ablated, the softmax redistributes weights primarily to positions with the highest remaining scores. If these positions contain values $V_j$ that contribute similar information, the output remains virtually unchanged.

Figure 1 illustrates this with attention weights $[0.5, 0.5, 0]$ and value vectors $[1, 1, 0]$. Note that the values are identical at the positions with non-zero attention weights. Individually ablating either attention score (i.e., softmax input) leaves the output unchanged, as the softmax shifts the weight to the other position with the same value. Only ablating both scores simultaneously affects the output, as the softmax shifts the attention weights to the position with a different value.

### 3.1 ATTENTION SELF-REPAIR RESULTS IN ZERO GRADIENT

Here we will prove that when the conditions for attention self-repair are met, the gradient of the model's final output logit $z$ with respect to the attention score $a_j$ will be near-zero. We can define the gradient as:

$$\frac{\partial z}{\partial a_j} = s_j \left( \frac{\partial z}{\partial s_j}(1 - s_j) - \sum_{k \neq j} \frac{\partial z}{\partial s_k} s_k \right) \tag{3}$$

where $\frac{\partial z}{\partial s_j}$ is the gradient of the output logit with respect to the attention weight $s_j$. This is computed as follows:

$$\frac{\partial z}{\partial s_j} = \frac{\partial z}{\partial o} \cdot V_j \tag{4}$$

To demonstrate the effect of attention self-repair on Equation (3), we first define a set of positions with substantial attention weights $\mathcal{I}_\epsilon = \{k : s_k > \epsilon\}$ where $\epsilon$ is close to zero. Recall that for self-repair to occur, the values $V_j$ at the positions with substantial attention weights must contribute similarly to the output. We can measure a value's unweighted contribution to the output with the dot product between the value and its upstream gradient (GradientXInput). This is equivalent to $\frac{\partial z}{\partial s_j}$ in Equation (4).

Therefore, the key condition for self-repair, which occurs when values at positions with substantial attention weights have similar contributions to the output, can be expressed mathematically as:

$$\frac{\partial z}{\partial s_j} \approx \frac{\partial z}{\partial s_k} \approx c \quad \forall j, k \in \mathcal{I}_\epsilon \tag{5}$$

where $c$ is some constant. Next, we use the property that softmax weights sum to 1:

$$\sum_{k \in \mathcal{I}_\epsilon} s_k \approx 1 \Leftrightarrow 1 - s_j \approx \sum_{\substack{k \in \mathcal{I}_\epsilon \\ k \neq j}} s_k \tag{6}$$

Substituting these into Equation (3), we get:

$$\frac{\partial z}{\partial a_j} \approx s_j \big( c \sum_{\substack{k \in \mathcal{I}_\epsilon \\ k \neq j}} s_k - \sum_{\substack{k \in \mathcal{I}_\epsilon \\ k \neq j}} c s_k \big) = s_j c \big( \sum_{\substack{k \in \mathcal{I}_\epsilon \\ k \neq j}} s_k - \sum_{\substack{k \in \mathcal{I}_\epsilon \\ k \neq j}} s_k \big) = 0 \tag{7}$$

This result reveals that standard gradient-based explanation methods underestimate the importance of attention scores when attention self-repair conditions are met. Even when an attention score $a_j$ significantly impacts the model's prediction, its gradient can be near-zero. This mathematical insight motivates our approach to modify gradient calculations to better reflect the true importance of components.

## 4 METHODS

To address self-repair and other gradient distortion issues in transformer models, we introduce *gradient interaction modification* (GIM), a comprehensive approach combining three complementary modifications of standard back-propagation. This section describes the three modifications: *Temperature-adjusted softmax gradients* (TSG), *Layernorm freeze*, and *Grad norm*.

### 4.1 TEMPERATURE-ADJUSTED SOFTMAX GRADIENTS

TSG modifies backpropagation through attention to address the self-repair problem presented in Section 3.1. The motivation behind TSG is well explained through the "firing squad" analogy from Pearl (2009): two soldiers fire at a prisoner, and we want to measure the causal importance of each soldier for the prisoner's death. If we measure this by separately preventing each soldier from firing, we would incorrectly conclude that both are innocent, since either shot alone is fatal. To measure their true responsibility, we must simultaneously prevent both shooters from firing.

Attention self-repair exhibits similar OR-gate behavior to the firing squad example. The gradient of the output with respect to the attention scores measures the effect of perturbing each attention score separately. Instead, we designed TSG to approximate the effect of perturbing multiple attention scores simultaneously. We developed TSG empirically by evaluating which techniques led to the fastest and most accurate approximation of jointly ablating multiple attention scores, while also producing faithful explanations. We will later demonstrate the accuracy of this approximation.

During backpropagation, TSG recomputes the softmax with a higher temperature ($T > 1$) before computing the softmax gradients. This results in a more uniform distribution of attention weights: large attention weights decrease while small weights increase. This increases our set of positions with significant weights $\mathcal{I}_\epsilon = \{k : s_k > \epsilon\}$. Recall that for self-repair to occur, the gradients $\frac{\partial z}{\partial s_j}$ must be approximately uniform across all positions in $\mathcal{I}_\epsilon$. By expanding $\mathcal{I}_\epsilon$ to include more positions, we increase the probability of a value vector disrupting the uniformity condition. When this uniformity condition is broken, the two terms in Equation (3) no longer completely cancel each other out, allowing for non-zero gradients.

### 4.2 LAYERNORM FREEZE & GRADIENT NORM

Layernorm freeze addresses the self-repair effect in the layer normalization by treating the normalization parameters as constants during backpropagation Rushing & Nanda (2024). Specifically, during the forward pass, we save the normalization factor $\sigma$, and during the backward pass, we divide the upstream gradient by the normalization factor $\sigma$ instead of backpropagating through it. This naively prevents the self-repair effect by preventing layer normalization from changing due to perturbations.

Gradient norm normalizes the gradient after variable multiplications Achtibat et al. (2024). During backpropagation, it divides the gradient by the number of inputs involved in the multiplication. Achtibat et al. (2024) proved the optimality of this approach using Taylor decomposition and Shapley values. Specifically, we divide the gradient by 2 for three key multiplicative interactions in transformer models: 1) attention-value, 2) query-key, and 3) MLP gate-projection.

## 5 EXPERIMENTAL SETUP

Our experiments were designed around three questions: 1) Does attention self-repair occur in large language models? 2) Does GIM produce more faithful explanations than existing circuit identification and feature attribution methods? 3) How much does each of GIM's three modifications (TSG, layernorm freeze, and grad norm) contribute to the increased faithfulness? This section describes how we ran the experiments.

### 5.1 MODELS, DATASETS, AND EVALUATION METRICS

For our experiments, we use seven instruction-tuned large language models spanning three model families of various sizes: Gemma-2 (2B, 9B), LLaMA-3.2 (1B, 3B), LLaMA-3.1 (8B), and Qwen-2.5 (1.5B, 3B) Team et al. (2024); Grattafiori et al. (2024); Qwen et al. (2025).

We use six datasets spanning four tasks: BoolQ (question-answering), FEVER and SciFact (fact verification), Twitter sentiment classification and Movie review (sentiment classification), and HateXplain (hate speech detection). These datasets were selected based on their diversity and prevalence in prior attribution studies Lyu et al. (2024). We provide more details on the datasets in Section A.3.

We evaluate the faithfulness of the attributions using Comprehensiveness and Sufficiency DeYoung et al. (2020).

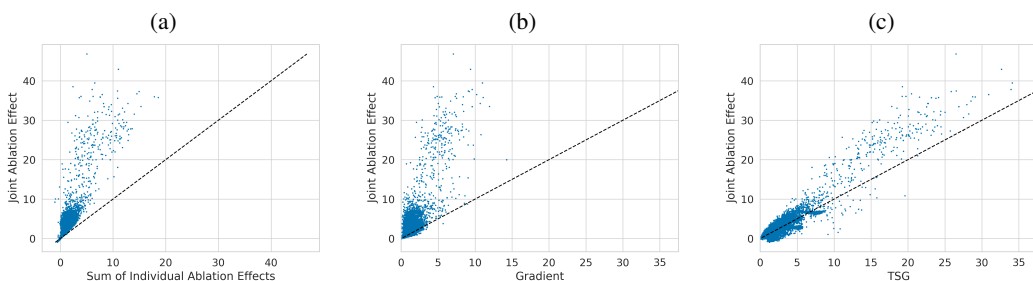

Figure 2: The attention self-repair effect and how temperature-adjusted softmax gradients approximate the joint ablation effect for LLAMA-3.2 1B on FEVER. The figures compare ablating the two largest attention scores jointly with a) ablating them separately, b) gradients, and c) TSG.

**Comprehensiveness** ($\uparrow$ the higher the better) measures the average output change after cumulatively ablating top-attributed features:

$$\text{Comp}(f, \boldsymbol{x}, \boldsymbol{e}) = \frac{1}{Nf(\boldsymbol{x})} \sum_{i=1}^{N} f(\boldsymbol{x}) - f(p(\boldsymbol{x}, \text{rank}(\boldsymbol{e})_{1:i})) \qquad (8)$$

where $f$ is the model, $\boldsymbol{x}$ is either the input embeddings or hidden representation, $\boldsymbol{e}$ contains attribution scores, $N$ is the number of input features, $p$ is the perturbation function, and $\text{rank}(\cdot)$ orders features by attribution.

**Sufficiency** ($\downarrow$ the lower the better) measures the average output change after cumulatively ablating the lowest-attributed features:

$$\text{Suff}(f, \boldsymbol{x}, \boldsymbol{e}) = \frac{1}{Nf(\boldsymbol{x})} \sum_{i=1}^{N} f(\boldsymbol{x}) - f(p(\boldsymbol{x}, \text{rank}(-\boldsymbol{e})_{1:i})) \qquad (9)$$

While Comprehensiveness is usually computed on the input layer, it can be computed at any layer.

## 5.2 EXPERIMENTS

**Attention self-repair in large language models:** We investigate whether the attention self-repair problem occurs in large language models and evaluate the effectiveness of TSG in approximating the effect of perturbing multiple attention scores simultaneously.

To identify instances of self-repair in practice, we apply the following procedure. We focus on the top 1% most important attention weights, which we compute as $\frac{\partial z}{\partial s_j} s_j$. From these, we select only attention weight vectors that have multiple significant attention weights (each $> 0.01$). Finally, we classify these cases as exhibiting self-repair if the value vectors at positions with significant attention weights are similar to each other. To identify similar value vectors, we first compute their contributions to the model output: $\frac{\partial z}{\partial o} V_j$. We then calculate the coefficient of variation (standard deviation divided by mean) across these contributions. Intuitively, a low coefficient of variation indicates that the value vectors contribute similarly to the output, which is the condition under which self-repair occurs, and gradients cancel out. If this value is less than $0.1$, we classify the case as self-repair.

For each identified self-repair in attention, we measure the joint effect by ablating the two largest attention scores simultaneously versus the sum of their individual ablation effects. If the joint effect is larger than the sum of the individual effects, the attention scores exhibit OR-gate-like behaviour, which is equivalent to self-repair Tsang et al. (2020). We quantify these effects by the change in weighted gradient sum $\sum_k \frac{\partial z}{\partial s_k} s_k$. We do this instead of computing the full forward pass to avoid other components compensating for the ablations. Finally, we show that TSG better approximates the joint effect than standard gradients by comparing both with the joint effects.

**Faithfulness of GIM:** For feature attribution evaluation, we compare GIM with five gradient-based methods: GradientXInput (GxI) Sundararajan et al. (2017), Integrated gradients (IG) Sundararajan et al. (2017), DeepLIFT Shrikumar et al. (2017a), TransformerLRP Ali et al. (2022), and

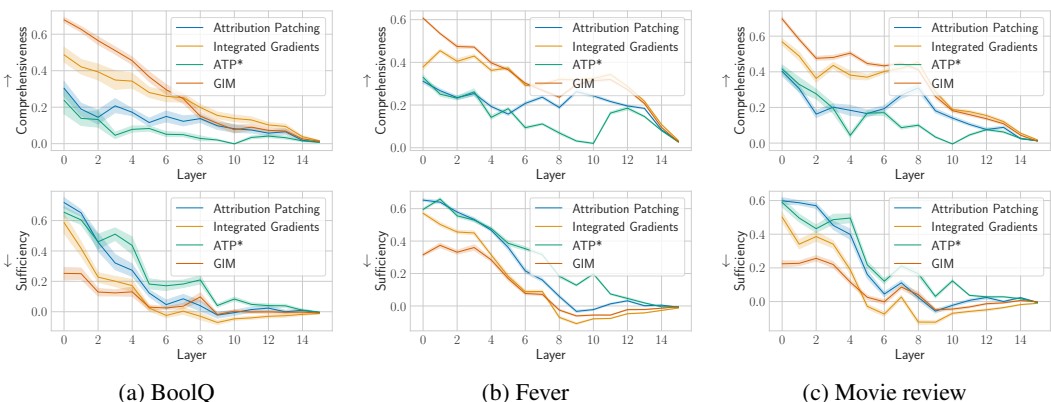

(a) BoolQ          (b) Fever          (c) Movie review

Figure 3: Comparison of circuit identification methods for LLAMA-3.2 1B (95% CI). The top row depicts comprehensiveness per layer (↑). The bottom row depicts sufficiency (↓).

AttnLRP Hanna et al. (2024). We did not evaluate perturbation-based methods because they are too computationally expensive for large language models on long inputs Syed et al. (2023). We computed comprehensiveness and sufficiency by replacing tokens with the token representing a space.

For circuit identification evaluation, we compare GIM with attribution patching (ATP) Syed et al. (2023), ATP*[2] Kramár et al. (2024a), and Integrated Gradients Hanna et al. (2024) by computing comprehensiveness and sufficiency at each layer separately, by replacing the representations with the layer-average representations. We also evaluate GIM on the Mechanistic Interpretability Benchmarks Mueller et al. (2025). We provide more details in Section A.1.

All attribution scores follow the formula $(\boldsymbol{x} - \tilde{\boldsymbol{x}}) \cdot \boldsymbol{\alpha}$, where $\boldsymbol{x}$ and $\tilde{\boldsymbol{x}}$ are original and counterfactual representations, and $\boldsymbol{\alpha}$ represents method-specific attributions. For the feature attribution evaluation, we used the token representing a space as the counterfactuals. For the circuit identification evaluation, we used the layer-average representations as counterfactuals. We used a temperature of 2 for TSG, which we chose based on the results on Gemma-2 2B on the FEVER and HateXplain datasets.

**Identifying the importance of each GIM modification:** Our final experiment is an ablation study, where we estimate the importance of the three modifications in GIM. We measure the change of feature attribution faithfulness when cumulatively adding modifications to the standard GradientXInput.

## 6 RESULTS

**Self-repair is prevalent in attention mechanisms:** Figure 2a compares the effect of ablating the two largest attention scores simultaneously (y-axis) versus the sum of their individual ablation effects (x-axis) for the samples we identified as self-repair. The consistent gap above the diagonal provides clear empirical evidence of self-repair: individual attention scores show disproportionately small effects when ablated separately. These results empirically validate our theoretical analysis from Section 3.1. We provide results for other models and datasets in Section A.10. On average, we identified self-repair in the model 65–1200 times per input, depending on the model and dataset.

**TSG accurately approximates the joint effect:** Figure 2b and 2c compare standard gradient attribution and TSG with the joint effect. As intended, TSG more accurately approximates the effect of perturbing multiple attention scores simultaneously. We provide similar results for other models and datasets in Section A.10.

**GIM is more faithful than other gradient-based circuit identification methods:** Figure 3 shows comprehensiveness and sufficiency metrics across layers for LLAMA-3.2 1B. GIM significantly outperforms other circuit identification methods in early layers, while Integrated Gradients demonstrates slight advantages in middle and late layers. Similar patterns appear across other models and datasets (Section A.9). Csordás et al. (2025) showed that LLMs primarily use the attention heads in the early

---

[2]We only implemented ATP*'s softmax gradient fix component.

Table 1: Comparison of the faithfulness of gradient-based feature attribution methods. The best scores for each model-dataset pair are shown in bold.

| | | Comprehensiveness ↑ | | | | | | | Sufficiency ↓ | | | | | | |
| | | Gemma | | LLAMA | | | Qwen | | Gemma | | LLAMA | | | Qwen | |
| | | 2B | 9B | 1B | 3B | 8B | 1.5B | 3B | 2B | 9B | 1B | 3B | 8B | 1.5B | 3B |
|---|---|---|---|---|---|---|---|---|---|---|---|---|---|---|---|
| BoolQ | GxI | 0.09 (0.08) | 0.00 (0.02) | 0.18 (0.07) | 0.45 (0.07) | 0.39 (0.07) | 0.27 (0.13) | 0.57 (0.18) | 0.60 (0.09) | 0.43 (0.12) | 0.71 (0.07) | 0.39 (0.07) | 0.67 (0.06) | 0.54 (0.17) | 0.43 (0.11) |
| | IG | 0.51 (0.08) | 0.20 (0.12) | 0.20 (0.09) | 0.52 (0.07) | 0.40 (0.06) | -0.00 (0.00) | 0.48 (0.22) | 0.11 (0.04) | 0.12 (0.11) | 0.43 (0.11) | 0.34 (0.03) | 0.58 (0.07) | 0.63 (0.03) | 0.54 (0.22) |
| | DeepLIFT | 0.41 (0.15) | 0.15 (0.07) | 0.26 (0.09) | 0.36 (0.07) | 0.31 (0.08) | 0.20 (0.17) | 0.34 (0.12) | 0.37 (0.14) | 0.37 (0.13) | 0.58 (0.11) | 0.55 (0.09) | 0.74 (0.06) | 0.67 (0.14) | 0.77 (0.18) |
| | TransformerLRP | 0.32 (0.10) | 0.09 (0.08) | 0.57 (0.04) | 0.55 (0.09) | 0.17 (0.13) | 0.26 (0.19) | 0.27 (0.12) | 0.37 (0.23) | 0.07 (0.04) | 0.30 (0.05) | 0.30 (0.09) | 0.68 (0.10) | 0.49 (0.17) | 0.64 (0.05) |
| | AttnLRP | - | - | 0.66 (0.03) | 0.70 (0.02) | **0.61 (0.04)** | 0.67 (0.11) | **0.65 (0.03)** | — | — | 0.25 (0.06) | 0.17 (0.05) | 0.50 (0.05) | 0.15 (0.05) | 0.34 (0.11) |
| | GIM | **0.59 (0.02)** | **0.43 (0.04)** | **0.69 (0.03)** | **0.72 (0.03)** | 0.60 (0.05) | **0.68 (0.05)** | 0.61 (0.03) | **0.03 (0.04)** | **0.04 (0.03)** | **0.22 (0.05)** | **0.10 (0.04)** | **0.49 (0.04)** | **0.09 (0.04)** | **0.23 (0.10)** |
| FEVER | GxI | 0.03 (0.19) | 0.06 (0.08) | 0.39 (0.07) | 0.53 (0.12) | 0.46 (0.06) | 0.43 (0.21) | 0.47 (0.13) | 0.47 (0.11) | 0.40 (0.10) | 0.56 (0.07) | 0.63 (0.09) | 0.71 (0.04) | 0.59 (0.06) | 0.73 (0.10) |
| | IG | **0.47 (0.10)** | 0.32 (0.08) | 0.28 (0.08) | 0.51 (0.11) | 0.50 (0.10) | 0.03 (0.09) | 0.54 (0.14) | 0.02 (0.21) | 0.20 (0.08) | 0.51 (0.07) | 0.59 (0.09) | 0.62 (0.05) | 0.67 (0.03) | 0.61 (0.12) |
| | DeepLIFT | 0.12 (0.19) | 0.20 (0.10) | 0.33 (0.07) | 0.47 (0.07) | 0.53 (0.06) | 0.31 (0.10) | 0.43 (0.14) | 0.43 (0.13) | 0.36 (0.12) | 0.58 (0.07) | 0.65 (0.07) | 0.76 (0.05) | 0.64 (0.10) | 0.76 (0.10) |
| | TransformerLRP | 0.17 (0.19) | 0.22 (0.07) | 0.54 (0.06) | 0.68 (0.05) | 0.21 (0.07) | 0.30 (0.08) | 0.49 (0.08) | 0.44 (0.10) | 0.21 (0.04) | 0.32 (0.06) | 0.44 (0.09) | 0.52 (0.03) | 0.23 (0.08) | 0.49 (0.08) |
| | AttnLRP | - | - | 0.60 (0.03) | **0.75 (0.03)** | 0.65 (0.04) | **0.61 (0.04)** | 0.68 (0.03) | - | - | 0.32 (0.06) | 0.44 (0.09) | 0.52 (0.03) | **0.23 (0.08)** | 0.49 (0.08) |
| | GIM | 0.42 (0.11) | **0.43 (0.05)** | **0.62 (0.03)** | **0.75 (0.02)** | **0.69 (0.03)** | 0.51 (0.03) | **0.68 (0.04)** | **-0.01 (0.19)** | **0.09 (0.07)** | **0.26 (0.07)** | **0.39 (0.11)** | **0.51 (0.03)** | 0.31 (0.09) | **0.41 (0.08)** |
| HateXplain | GxI | 0.07 (0.05) | 0.08 (0.05) | 0.53 (0.08) | 0.38 (0.14) | 0.44 (0.08) | 0.49 (0.29) | 0.53 (0.15) | 0.59 (0.04) | 0.42 (0.06) | 0.60 (0.06) | 0.67 (0.07) | 0.73 (0.03) | 0.63 (0.09) | 0.63 (0.10) |
| | IG | 0.62 (0.03) | 0.37 (0.09) | 0.49 (0.08) | 0.64 (0.09) | 0.44 (0.05) | 0.00 (0.01) | 0.46 (0.11) | 0.35 (0.03) | 0.32 (0.07) | 0.58 (0.06) | 0.56 (0.06) | 0.61 (0.05) | 0.64 (0.03) | 0.72 (0.14) |
| | DeepLIFT | 0.36 (0.05) | 0.29 (0.05) | 0.55 (0.07) | 0.36 (0.15) | 0.41 (0.09) | 0.29 (0.08) | 0.42 (0.11) | 0.52 (0.04) | 0.45 (0.05) | 0.59 (0.06) | 0.70 (0.08) | 0.73 (0.03) | 0.74 (0.05) | 0.73 (0.09) |
| | TransformerLRP | 0.54 (0.03) | 0.14 (0.04) | 0.51 (0.09) | 0.58 (0.10) | 0.33 (0.08) | 0.20 (0.08) | 0.30 (0.08) | 0.38 (0.06) | 0.24 (0.03) | 0.59 (0.07) | 0.57 (0.05) | 0.67 (0.03) | 0.66 (0.06) | 0.71 (0.07) |
| | AttnLRP | - | - | **0.68 (0.04)** | **0.70 (0.04)** | 0.60 (0.02) | **0.85 (0.06)** | 0.59 (0.09) | - | - | 0.30 (0.05) | 0.47 (0.04) | **0.42 (0.04)** | **0.33 (0.05)** | 0.71 (0.07) |
| | GIM | **0.64 (0.01)** | **0.48 (0.02)** | **0.68 (0.04)** | 0.64 (0.08) | **0.65 (0.02)** | 0.80 (0.06) | **0.63 (0.08)** | **0.26 (0.04)** | **0.14 (0.05)** | **0.26 (0.06)** | **0.46 (0.04)** | 0.47 (0.06) | 0.35 (0.06) | **0.51 (0.15)** |
| Movie | GxI | 0.15 (0.13) | 0.04 (0.09) | 0.34 (0.10) | 0.51 (0.10) | 0.34 (0.11) | 0.42 (0.06) | 0.52 (0.10) | 0.61 (0.10) | 0.52 (0.10) | 0.61 (0.09) | 0.51 (0.07) | 0.71 (0.03) | 0.68 (0.12) | 0.76 (0.17) |
| | IG | **0.60 (0.06)** | 0.38 (0.08) | 0.28 (0.12) | 0.69 (0.08) | 0.41 (0.08) | 0.00 (0.01) | 0.58 (0.21) | 0.17 (0.06) | 0.13 (0.08) | 0.49 (0.09) | 0.36 (0.11) | 0.52 (0.07) | 0.71 (0.03) | 0.56 (0.16) |
| | DeepLIFT | 0.44 (0.11) | 0.22 (0.10) | 0.31 (0.10) | 0.62 (0.09) | 0.48 (0.07) | 0.49 (0.12) | 0.35 (0.14) | 0.35 (0.13) | 0.48 (0.07) | 0.60 (0.08) | 0.49 (0.07) | 0.71 (0.06) | 0.56 (0.09) | 0.77 (0.08) |
| | TransformerLRP | 0.48 (0.08) | 0.26 (0.05) | 0.61 (0.08) | 0.70 (0.04) | 0.22 (0.05) | 0.23 (0.12) | 0.49 (0.23) | 0.27 (0.12) | 0.13 (0.07) | 0.35 (0.11) | 0.40 (0.04) | 0.83 (0.04) | 0.74 (0.16) | 0.67 (0.08) |
| | AttnLRP | - | - | 0.69 (0.04) | 0.76 (0.02) | **0.72 (0.04)** | 0.78 (0.05) | **0.77 (0.11)** | - | - | 0.25 (0.06) | 0.30 (0.05) | 0.48 (0.03) | 0.19 (0.06) | 0.46 (0.16) |
| | GIM | **0.60 (0.03)** | **0.44 (0.03)** | **0.71 (0.04)** | **0.78 (0.02)** | 0.71 (0.02) | **0.83 (0.03)** | 0.76 (0.09) | **0.11 (0.06)** | **0.07 (0.04)** | **0.19 (0.06)** | **0.30 (0.05)** | **0.48 (0.03)** | **0.17 (0.05)** | **0.31 (0.19)** |
| SciFact | GxI | 0.09 (0.19) | 0.05 (0.12) | 0.27 (0.07) | 0.42 (0.15) | 0.40 (0.08) | 0.34 (0.20) | 0.57 (0.20) | 0.51 (0.11) | 0.39 (0.13) | 0.57 (0.09) | 0.59 (0.09) | 0.72 (0.06) | 0.56 (0.11) | 0.63 (0.16) |
| | IG | **0.58 (0.14)** | 0.32 (0.14) | 0.27 (0.09) | 0.46 (0.14) | 0.50 (0.08) | 0.01 (0.02) | 0.65 (0.19) | 0.18 (0.19) | 0.20 (0.18) | 0.46 (0.08) | 0.56 (0.11) | 0.57 (0.07) | 0.69 (0.03) | 0.54 (0.14) |
| | DeepLIFT | 0.28 (0.25) | 0.27 (0.17) | 0.28 (0.07) | 0.43 (0.13) | 0.50 (0.09) | 0.25 (0.15) | 0.62 (0.22) | 0.47 (0.12) | 0.37 (0.16) | 0.58 (0.08) | 0.61 (0.08) | 0.75 (0.07) | 0.69 (0.16) | 0.65 (0.14) |
| | TransformerLRP | 0.30 (0.22) | 0.22 (0.12) | 0.56 (0.06) | 0.56 (0.12) | 0.15 (0.11) | 0.39 (0.15) | 0.51 (0.12) | 0.41 (0.19) | 0.22 (0.13) | 0.33 (0.08) | 0.43 (0.13) | 0.76 (0.05) | 0.50 (0.14) | 0.67 (0.14) |
| | AttnLRP | - | - | 0.64 (0.03) | 0.71 (0.03) | 0.65 (0.04) | **0.67 (0.13)** | **0.74 (0.10)** | - | - | 0.30 (0.07) | 0.35 (0.10) | 0.49 (0.07) | **0.22 (0.08)** | 0.40 (0.12) |
| | GIM | 0.57 (0.08) | **0.40 (0.08)** | **0.67 (0.03)** | **0.74 (0.04)** | **0.67 (0.04)** | 0.62 (0.08) | **0.74 (0.06)** | **0.04 (0.18)** | **0.08 (0.13)** | **0.22 (0.08)** | **0.31 (0.09)** | **0.49 (0.08)** | 0.26 (0.07) | **0.38 (0.11)** |
| Twitter | GxI | 0.40 (0.09) | 0.27 (0.06) | 0.41 (0.07) | 0.48 (0.10) | 0.45 (0.05) | 0.48 (0.07) | 0.37 (0.06) | 0.51 (0.07) | 0.37 (0.06) | 0.64 (0.05) | 0.75 (0.05) | 0.77 (0.03) | 0.73 (0.06) | 0.64 (0.04) |
| | IG | **0.53 (0.06)** | 0.34 (0.06) | 0.41 (0.08) | 0.70 (0.07) | 0.55 (0.05) | 0.40 (0.05) | 0.67 (0.13) | 0.36 (0.06) | 0.33 (0.05) | 0.62 (0.05) | **0.59 (0.05)** | 0.66 (0.04) | 0.71 (0.09) | 0.66 (0.07) |
| | DeepLIFT | 0.44 (0.06) | 0.23 (0.06) | 0.42 (0.06) | 0.65 (0.06) | 0.54 (0.05) | 0.51 (0.07) | **0.75 (0.12)** | 0.54 (0.05) | 0.44 (0.05) | 0.62 (0.05) | 0.67 (0.03) | 0.75 (0.04) | 0.75 (0.04) | 0.72 (0.05) |
| | TransformerLRP | 0.43 (0.05) | 0.26 (0.05) | 0.58 (0.05) | 0.62 (0.04) | 0.34 (0.06) | 0.41 (0.08) | 0.50 (0.11) | 0.43 (0.05) | 0.38 (0.04) | 0.49 (0.04) | 0.66 (0.03) | 0.76 (0.03) | 0.73 (0.04) | 0.72 (0.05) |
| | AttnLRP | - | - | 0.64 (0.04) | **0.73 (0.03)** | 0.68 (0.02) | 0.75 (0.07) | 0.66 (0.05) | - | - | 0.42 (0.04) | 0.61 (0.04) | 0.57 (0.03) | **0.48 (0.04)** | 0.66 (0.08) |
| | GIM | 0.52 (0.06) | **0.38 (0.05)** | **0.68 (0.03)** | 0.72 (0.03) | **0.69 (0.02)** | **0.76 (0.06)** | 0.72 (0.06) | **0.34 (0.04)** | **0.22 (0.03)** | **0.41 (0.04)** | 0.61 (0.05) | **0.56 (0.03)** | 0.55 (0.04) | **0.58 (0.04)** |

layers. Since most of our modifications target the attention heads, this could explain the pattern. Across all methods, the scores worsen in the top layers of the LLM. This could either be because they become less faithful or because the upper and lower limits of comprehensiveness and sufficiency change across layers Edin et al. (2025). In Section A.1, we show that GIM outperforms the other circuit discovery methods on the Mechanistic Interpretability Benchmark.

**GIM is often more faithful than other gradient-based feature attribution methods:** Table 1 compares GIM against five other gradient-based attribution methods across multiple datasets and model architectures. GIM significantly outperforms traditional methods, such as GradientXInput, Integrated Gradients, and DeepLIFT, in nearly all cases. While AttnLRP outperforms GIM on several dataset-model pairs, GIM achieves the highest comprehensiveness and sufficiency scores for the majority of combinations, establishing it as the most consistently faithful method overall.

**All modifications contribute to GIM's faithfulness improvements:** Figure 4 compares how adding different GIM modifications to GradientXInput improves feature attribution faithfulness. Each point represents the average improvement for a dataset compared to the GradientXInput baseline. Blue boxplots show feature attribution performance with LayerNorm freeze + Gradient Normalization, while orange boxplots include TSG as well (full GIM). The results demonstrate that LayerNorm freeze + Gradient Normalization substantially improves both attribution metrics across all models. Adding TSG further enhances attribution quality for most models, with particularly strong improvements for Gemma-2 2B and LLAMA models. However, TSG appears less beneficial for Qwen-2.5 1.5B, where it occasionally reduces attribution faithfulness. Qwen-2.5 1.5B had a hundred-fold larger gradients than the other models, which perhaps is problematic for TSG (see Figure 11). While this inter-model variance is undesirable, it is even larger for established methods such as IG and GxI. Comprehensive feature attribution results for all modification combinations are available in Table 6.

## 7 DISCUSSION

**Should gradients estimate the joint effect?** We designed TSG to approximate the effect of perturbing multiple attention scores simultaneously to address the self-repair problem. However, is this always desirable? Whether to approximate the effect of perturbing one or multiple attention scores depends on the causal question being asked. If a perturbation would cause changes in multiple interacting attention scores, then we should estimate the joint effect; otherwise, we should estimate the individual effect. However, when perturbing early-layer variables, determining which attention scores in deep layers are affected becomes difficult because the number of potential causal paths grows exponentially with depth. We naively assumed that perturbations would always impact all interacting attention

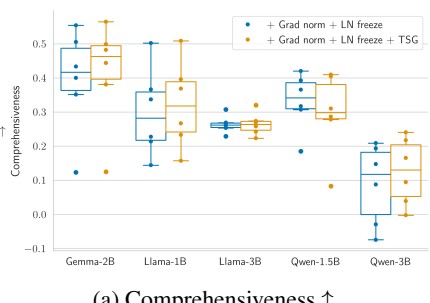 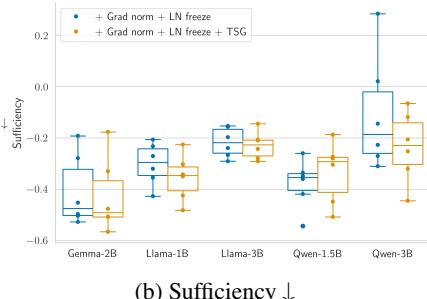

(a) Comprehensiveness ↑          (b) Sufficiency ↓

Figure 4: Relative improvement of adding modifications to GradientXInput. We compare adding layernorm freeze and grad norm with also adding TSG. Each point represents the average statistics of a dataset. The y-axis shows the improvements over GradientXInput.

scores. This assumption may fail for certain inputs, models, and types of perturbations, which could explain why GIM underperforms other methods for Qwen 1.5B in Table 1 and in the late layers in Figure 3. TSG is a naive solution to the self-repair problem; future work should investigate methods to estimate whether a perturbation will cause changes in multiple interacting attention scores, enabling a more principled selection between joint and individual effect estimation.

**Could GIM improve model training?** Since GIM's gradient modifications more accurately attribute importance to model components during interpretation, we speculate that they might also improve model training itself. By implementing these modifications during backpropagation, we might achieve more precise parameter updates that account for feature interactions and self-repair mechanisms. We suggest that future work empirically evaluate whether models trained with GIM-modified gradients demonstrate faster convergence or better performance compared to standard training approaches.

**Limitations:** While our experiments demonstrate that TSG improves the faithfulness of the attributions and accurately approximates the effect of joint ablation during self-repair, we cannot conclusively establish a causal link between these observations. Temperature adjustment affects backpropagation in multiple ways beyond mitigating self-repair, and these additional effects may contribute to the improved faithfulness we observe. Furthermore, we did not find an explanation for why TSG improved faithfulness across all models except for Qwen-2.5 1.5B. We observed that its gradient was a hundred-fold larger than the other models', but could not provide any definite evidence that this caused the decreased faithfulness.

## 8 CONCLUSION

We identified attention self-repair, a phenomenon where softmax normalization redistributes weights during perturbation, causing interpretability methods to underestimate component importance. Our proposed method, GIM, addresses this through TSG, while also incorporating layer normalization freeze and gradient normalization to handle other issues in transformer interpretability. Our evaluations across multiple models and datasets show that GIM consistently outperforms existing circuit identification and feature attribution methods in faithfulness. By accounting for self-repair during backpropagation, GIM more accurately quantifies the importance of model components, thereby advancing our understanding of neural networks and enhancing our ability to interpret them.

## 9 REPRODUCIBILITY STATEMENT

Our code is available at `https://anonymous.4open.science/r/explainable_transformer-D693/README.md`. In this code repository, we provide commands for running every experiment. In addition to the code, we provide experimental details in Section 5.2. We also provide information about the required licenses for using the datasets in Section A.4, and the computational resources we used for the experiments in Section A.4.

ACKNOWLEDGEMENT

Thank you, Jakob Havtorn, Lasse Borgholt, and Søren Hauberg, for discussions and advice. This research was partially funded by the Innovation Fund Denmark via the Industrial Ph.D. Program (grant no. 2050-00040B) and Academy of Finland (grant no. 322653).

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

# A    APPENDIX / SUPPLEMENTAL MATERIAL

## A.1    MECHANISTIC INTERPRETABILITY BENCHMARK

We evaluate GIM on the Circuit Location Track on the Mechanistic Interpretability Benchmark on the Mueller et al. (2025). This benchmark evaluates the faithfulness of circuits identified by circuit discovery methods. It covers multiple tasks: indirect object identification (IOI), Arithmetic, multiple-choice question answering (MCQA), and AI2 Reasoning Challenge (ARC). We compare GIM with the circuit discovery methods evaluated by Mueller et al. (2025). These methods include Edge Activation Patching (EActP) Conmy et al., Edge Attribution Patching (EAP) Syed et al. (2023), EAP with integrated gradients (EAP-IG) Hanna et al. (2024), Node Attribution Patching (NAP) Syed et al. (2023), Information Flow Routes (IFR) Ferrando & Voita (2024), and Uniform Gradient Sampling (UGS) Li & Janson (2024). EAP and EAP-IG are similar to ATP and Integrated Gradients, but modified to discover edges in model circuits. We use the integrated circuit performance ratio (CPR) and the integrated circuit model distance (CMD) metrics to compare the methods. Similar to Mueller et al. (2025), we used the absolute values of the attribution scores when computing the CMD scores. We evaluate on GPT-2 120M, Qwen-2.5 0.5B, Gemma-2 2B, and LLama-3.1 8B. We refer to Mueller et al. (2025) for more details about the datasets, metrics, and circuit discovery methods.

Table 2 compares circuit discovery methods using the CPR metric. We copied the results of the other methods from the original paper Mueller et al. (2025). GIM significantly outperforms the other methods on most tasks and models.

Table 3 compares circuit discovery methods using the CMD metric. Here, GIM performs comparably to EAP-IG. This difference in relative performance between metrics can be explained by GIM's superior ability to distinguish between negative and positive attributions. Since CMD uses absolute values of attribution scores while CPR preserves the sign, GIM's strength in capturing attribution polarity translates directly into improved CPR performance but offers less advantage for CMD.

Table 2: **CPR** scores across circuit localization methods and ablation types. All evaluations were performed using counterfactual ablations. Higher scores are better. Arithmetic scores are averaged across addition and subtraction. We **bold** and underline the best and second-best methods per column, respectively. We copied the results of the other methods from the original paper Mueller et al. (2025).

| Method | IOI | | | | Arithmetic | MCQA | | | ARC (E) | | ARC (C) |
|---|---|---|---|---|---|---|---|---|---|---|---|
| | GPT-2 | Qwen-2.5 | Gemma-2 | Llama-3.1 | Llama-3.1 | Qwen-2.5 | Gemma-2 | Llama-3.1 | Gemma-2 | Llama-3.1 | Llama-3.1 |
| Random | 0.25 | 0.28 | 0.30 | 0.25 | 0.25 | 0.27 | 0.32 | 0.26 | 0.32 | 0.26 | 0.25 |
| EActP (CF) | **2.30** | 1.21 | - | - | - | 0.85 | - | - | - | - | - |
| EAP (mean) | 0.29 | 0.71 | 0.68 | 0.98 | 0.35 | 0.29 | 0.33 | 0.13 | 0.26 | 0.34 | 0.80 |
| EAP (CF) | 1.20 | 0.26 | 1.29 | 0.85 | 0.55 | 0.85 | 1.49 | 1.00 | 1.08 | 0.80 | 0.82 |
| EAP (OA) | 0.95 | 0.70 | - | - | - | 0.29 | - | - | - | - | - |
| EAP-IG-inputs (CF) | 1.85 | 1.63 | 3.20 | 2.08 | 0.99 | 1.16 | 1.64 | 1.05 | 1.53 | 1.04 | 0.98 |
| EAP-IG-activations (CF) | 1.82 | 1.63 | 2.07 | 1.60 | 0.98 | 0.77 | 1.57 | 0.79 | 1.70 | 0.71 | 0.63 |
| NAP (CF) | 0.28 | 0.30 | 0.30 | 0.26 | 0.27 | 0.38 | 1.47 | 1.69 | 1.01 | 0.26 | 0.26 |
| NAP-IG (CF) | 0.76 | 0.29 | 1.52 | 0.42 | 0.39 | 0.77 | 1.71 | **1.87** | 1.53 | 0.26 | 0.26 |
| IFR | 0.58 | 0.31 | 0.25 | 0.09 | 0.89 | 0.40 | 0.38 | 0.52 | 0.34 | 0.36 | 0.24 |
| UGS | 0.97 | 0.98 | - | - | - | 1.17 | - | - | - | - | - |
| GIM (CF) | 2.24 | **2.02** | **3.54** | **2.54** | **1.09** | **2.47** | **2.52** | 1.65 | **2.36** | **1.39** | **1.65** |

## A.2    ABLATION STUDY

Table 4 shows the impact of ablating each modification (TSG, Layernorm freeze, and Grad norm). Results that are worse than the standard GIM are shown in bold and underscored if they are significant. The table shows that ablating each modification deteriorates performance for most model-dataset pairs, indicating that they are important.

For some model-dataset pairs, ablating a modification may improve performance, indicating that the modification actually worsens it. However, variance across models and datasets is expected. In the explainability community, Integrated Gradients (IG) is recognized as superior to GradientXInput (GXI) Sundararajan et al. (2017); Li et al. (2023). Yet, in Table 5, we show that while IG often outperforms GXI, the results are inconsistent across models and datasets.

Table 3: **CMD** scores across circuit localization methods and ablation types (higher is better). All evaluations were performed using counterfactual ablations. Arithmetic scores are averaged across addition and subtraction. We **bold** and underline the best and second-best methods per column, respectively. We copied the results of the other methods from the original paper Mueller et al. (2025).

| Method | IOI | | | | Arithmetic | MCQA | | | ARC (E) | | ARC (C) |
|---|---|---|---|---|---|---|---|---|---|---|---|
| | GPT-2 | Qwen-2.5 | Gemma-2 | Llama-3.1 | Llama-3.1 | Qwen-2.5 | Gemma-2 | Llama-3.1 | Gemma-2 | Llama-3.1 | Llama-3.1 |
| Random | 0.75 | 0.72 | 0.69 | 0.74 | 0.75 | 0.73 | 0.68 | 0.74 | 0.68 | 0.74 | 0.74 |
| EActP (CF) | **0.02** | 0.49 | - | - | - | 0.36 | - | - | - | - | - |
| EAP (mean) | 0.29 | 0.18 | 0.25 | 0.04 | 0.07 | 0.21 | 0.20 | 0.16 | 0.22 | 0.28 | 0.20 |
| EAP (CF) | 0.03 | 0.15 | 0.06 | **0.01** | 0.01 | 0.07 | 0.08 | **0.09** | **0.04** | **0.11** | 0.18 |
| EAP (OA) | 0.30 | 0.16 | - | - | - | 0.11 | - | - | - | - | - |
| EAP-IG-inputs (CF) | 0.03 | 0.02 | 0.04 | **0.01** | **0.00** | 0.08 | 0.06 | 0.14 | **0.04** | **0.11** | 0.22 |
| EAP-IG-activations (CF) | 0.03 | **0.01** | **0.03** | **0.01** | **0.00** | **0.05** | 0.07 | 0.13 | **0.04** | 0.30 | 0.37 |
| NAP (CF) | 0.38 | 0.33 | 0.37 | 0.29 | 0.28 | 0.30 | 0.35 | 0.32 | 0.33 | 0.69 | 0.69 |
| NAP-IG (CF) | 0.27 | 0.20 | 0.26 | 0.19 | 0.18 | 0.18 | 0.29 | 0.33 | 0.28 | 0.67 | 0.67 |
| IFR | 0.42 | 0.69 | 0.75 | 0.83 | 0.22 | 0.60 | 0.62 | 0.48 | 0.66 | 0.64 | 0.76 |
| UGS | 0.03 | 0.03 | - | - | - | 0.20 | - | - | - | - | - |
| GIM (CF) | **0.02** | 0.02 | 0.05 | **0.01** | **0.00** | 0.10 | **0.03** | 0.21 | **0.04** | 0.16 | **0.17** |

Table 4: Performance when ablating each modification in GIM separately. Scores are shown as the mean (standard deviation). Average scores worse than GIM are shown in bold, while scores that are significantly worse are underlined (independent t-tests with $0.05 \leq \alpha$)

| | | Comprehensiveness ↑ | | | | | Sufficiency ↓ | | | | |
|---|---|---|---|---|---|---|---|---|---|---|---|
| | | Gemma | LLAMA | | Qwen | | Gemma | LLAMA | | Qwen | |
| | | 2B | 1B | 3B | 1.5B | 3B | 2B | 1B | 3B | 1.5B | 3B |
| BoolQ | GIM | 0.59 (0.02) | 0.69 (0.03) | 0.72 (0.03) | 0.68 (0.05) | 0.61 (0.03) | 0.03 (0.04) | 0.22 (0.05) | 0.10 (0.04) | 0.09 (0.04) | 0.23 (0.10) |
| | - LN freeze | **0.37 (0.14)** | **0.36 (0.09)** | **0.49 (0.14)** | **0.45 (0.07)** | 0.67 (0.07) | **0.46 (0.11)** | **0.57 (0.07)** | **0.58 (0.04)** | **0.68 (0.06)** | **0.39 (0.10)** |
| | - Grad norm | 0.09 (0.09) | **0.45 (0.07)** | **0.47 (0.07)** | **0.37 (0.29)** | **0.18 (0.10)** | **0.53 (0.11)** | **0.24 (0.05)** | **0.30 (0.12)** | **0.62 (0.14)** | **0.72 (0.19)** |
| | - TSG | 0.53 (0.05) | **0.68 (0.03)** | 0.72 (0.02) | 0.69 (0.07) | 0.66 (0.04) | **0.07 (0.05)** | **0.28 (0.04)** | **0.12 (0.04)** | **0.12 (0.05)** | **0.29 (0.10)** |
| FEVER | GIM | 0.42 (0.11) | 0.62 (0.03) | 0.75 (0.02) | 0.51 (0.03) | 0.68 (0.04) | -0.01 (0.19) | 0.26 (0.07) | 0.39 (0.11) | 0.31 (0.09) | 0.41 (0.08) |
| | - LN freeze | **0.30 (0.11)** | **0.34 (0.08)** | **0.61 (0.05)** | 0.54 (0.06) | 0.70 (0.05) | **0.32 (0.12)** | **0.55 (0.05)** | **0.72 (0.05)** | **0.49 (0.08)** | **0.49 (0.08)** |
| | - Grad norm | 0.10 (0.14) | **0.49 (0.08)** | **0.38 (0.10)** | **0.33 (0.17)** | **0.43 (0.15)** | **0.38 (0.13)** | **0.38 (0.09)** | **0.67 (0.09)** | **0.65 (0.07)** | **0.72 (0.12)** |
| | - TSG | **0.39 (0.13)** | **0.60 (0.03)** | 0.76 (0.03) | 0.62 (0.04) | **0.66 (0.04)** | -0.03 (0.21) | **0.29 (0.07)** | 0.39 (0.10) | 0.25 (0.08) | **0.46 (0.08)** |
| HateXplain | GIM | 0.64 (0.01) | 0.68 (0.04) | 0.64 (0.08) | 0.80 (0.06) | 0.63 (0.08) | 0.26 (0.04) | 0.26 (0.06) | 0.46 (0.04) | 0.35 (0.06) | 0.51 (0.15) |
| | - LN freeze | **0.46 (0.02)** | **0.61 (0.04)** | 0.64 (0.03) | **0.54 (0.07)** | 0.81 (0.05) | **0.51 (0.04)** | **0.45 (0.06)** | **0.58 (0.06)** | **0.73 (0.06)** | 0.45 (0.05) |
| | - Grad norm | 0.22 (0.05) | **0.57 (0.13)** | **0.28 (0.12)** | **0.17 (0.10)** | **0.15 (0.08)** | **0.56 (0.06)** | **0.47 (0.06)** | **0.70 (0.05)** | **0.76 (0.06)** | **0.80 (0.07)** |
| | - TSG | **0.63 (0.02)** | **0.67 (0.04)** | 0.64 (0.06) | 0.85 (0.09) | **0.50 (0.05)** | **0.31 (0.05)** | **0.37 (0.06)** | **0.52 (0.05)** | 0.28 (0.05) | **0.91 (0.11)** |
| Movie | GIM | 0.60 (0.03) | 0.71 (0.04) | 0.78 (0.02) | 0.83 (0.03) | 0.76 (0.09) | 0.11 (0.06) | 0.19 (0.06) | 0.30 (0.05) | 0.17 (0.05) | 0.31 (0.19) |
| | - LN freeze | **0.59 (0.07)** | **0.63 (0.05)** | **0.49 (0.05)** | **0.61 (0.07)** | 0.88 (0.06) | **0.49 (0.05)** | **0.45 (0.05)** | **0.67 (0.06)** | **0.75 (0.11)** | **0.43 (0.10)** |
| | - Grad norm | **0.27 (0.14)** | **0.62 (0.04)** | **0.62 (0.08)** | **0.37 (0.12)** | **0.35 (0.28)** | **0.56 (0.09)** | **0.25 (0.05)** | **0.35 (0.07)** | **1.10 (0.19)** | **0.76 (0.09)** |
| | - TSG | **0.55 (0.04)** | **0.68 (0.03)** | 0.78 (0.02) | **0.82 (0.04)** | **0.73 (0.12)** | **0.16 (0.06)** | **0.26 (0.05)** | **0.31 (0.05)** | 0.13 (0.05) | **0.45 (0.04)** |
| SciFact | GIM | 0.57 (0.08) | 0.67 (0.03) | 0.74 (0.04) | 0.62 (0.08) | 0.74 (0.06) | 0.04 (0.18) | 0.22 (0.08) | 0.31 (0.09) | 0.26 (0.07) | 0.38 (0.11) |
| | - LN freeze | **0.43 (0.09)** | **0.33 (0.09)** | **0.51 (0.08)** | **0.54 (0.07)** | 0.74 (0.10) | **0.43 (0.14)** | **0.53 (0.07)** | **0.68 (0.07)** | **0.75 (0.10)** | **0.42 (0.12)** |
| | - Grad norm | **0.23 (0.13)** | **0.53 (0.07)** | **0.34 (0.11)** | **0.28 (0.11)** | **0.36 (0.27)** | **0.48 (0.12)** | **0.33 (0.07)** | **0.59 (0.08)** | **0.74 (0.08)** | **0.76 (0.15)** |
| | - TSG | 0.60 (0.09) | **0.64 (0.03)** | 0.72 (0.04) | 0.66 (0.09) | **0.72 (0.07)** | **0.05 (0.18)** | **0.25 (0.07)** | 0.30 (0.09) | 0.20 (0.08) | **0.40 (0.12)** |
| Twitter | GIM | 0.52 (0.06) | 0.68 (0.03) | 0.72 (0.03) | 0.76 (0.06) | 0.72 (0.06) | 0.34 (0.04) | 0.41 (0.04) | 0.61 (0.05) | 0.55 (0.04) | 0.58 (0.04) |
| | - LN freeze | **0.47 (0.05)** | **0.48 (0.05)** | **0.55 (0.04)** | **0.63 (0.06)** | 0.80 (0.05) | **0.48 (0.04)** | **0.51 (0.04)** | **0.68 (0.02)** | **0.75 (0.04)** | **0.61 (0.04)** |
| | - Grad norm | **0.37 (0.07)** | **0.47 (0.06)** | **0.52 (0.09)** | **0.42 (0.06)** | **0.51 (0.14)** | **0.56 (0.05)** | **0.63 (0.05)** | **0.75 (0.04)** | **0.87 (0.05)** | **0.73 (0.05)** |
| | - TSG | 0.52 (0.06) | **0.64 (0.03)** | 0.73 (0.03) | 0.78 (0.06) | **0.65 (0.05)** | 0.32 (0.04) | **0.43 (0.04)** | 0.60 (0.03) | 0.47 (0.04) | **0.66 (0.08)** |

Table 6 shows the feature attribution faithfulness of each combination of modifications in GIM. The modifications have a much stronger effect when combined than when considered individually, especially when grad norm is combined with LN freeze.

## A.3 DATASETS

**BoolQ** comprises yes/no questions about Wikipedia articles, which is used in the SuperGLUE and ERASER benchmarks Clark et al. (2019); Wang et al.; DeYoung et al. (2020).

**FEVER** and **SciFact** are fact-verification datasets Thorne et al. (2018); Wadden et al. (2020). Fact verification is classifying whether a claim is true, given a document containing the necessary information. The documents and claims in FEVER are generated from Wikipedia, while SciFact comprises scientific abstracts and claims about COVID-19.

**Movie Reviews** and **Twitter Sentiment Extraction** are sentiment classification datasets, where the goal is to classify the sentiment of movie reviews and Twitter posts Zaidan et al. (2007); Maggie et al. (2020).

Table 5: Comparison of GradientXInput (GXI) and integrated gradients(IG). This table shows that, despite Integrated Gradient being an established explanation method, it is not consistently better than GradientXInput. Scores are shown as the mean (standard deviation). Scores where IG is better than GXI are shown in bold, the scores are underlined if IG is significantly better than GXI (independent t-tests with $0.05 \le \alpha$)

| | | Comprehensiveness ↑ | | | | | Sufficiency ↓ | | | | |
| | | Gemma | LLAMA | | Qwen | | Gemma | LLAMA | | Qwen | |
| | | 2B | 1B | 3B | 1.5B | 3B | 2B | 1B | 3B | 1.5B | 3B |
| BoolQ | GXI | 0.09 (0.08) | 0.18 (0.07) | 0.45 (0.07) | 0.27 (0.13) | 0.57 (0.18) | 0.60 (0.09) | 0.71 (0.07) | 0.39 (0.07) | 0.54 (0.17) | 0.43 (0.11) |
| | IG | **0.51 (0.08)** | **0.20 (0.09)** | **0.52 (0.07)** | -0.00 (0.00) | 0.48 (0.22) | **0.11 (0.04)** | **0.43 (0.11)** | **0.34 (0.03)** | 0.63 (0.03) | 0.54 (0.22) |
| FEVER | GXI | 0.03 (0.19) | 0.39 (0.07) | 0.53 (0.12) | 0.43 (0.21) | 0.47 (0.13) | 0.47 (0.11) | 0.56 (0.07) | 0.63 (0.09) | 0.59 (0.06) | 0.73 (0.10) |
| | IG | **0.47 (0.10)** | 0.28 (0.08) | 0.51 (0.11) | 0.03 (0.09) | **0.54 (0.14)** | **0.02 (0.21)** | **0.51 (0.07)** | **0.59 (0.09)** | 0.67 (0.03) | **0.61 (0.12)** |
| HateXplain | GXI | 0.07 (0.05) | 0.53 (0.08) | 0.38 (0.14) | 0.49 (0.29) | 0.53 (0.15) | 0.59 (0.04) | 0.60 (0.06) | 0.67 (0.07) | 0.63 (0.09) | 0.63 (0.10) |
| | IG | **0.62 (0.03)** | 0.49 (0.08) | **0.64 (0.09)** | 0.00 (0.01) | 0.46 (0.11) | **0.35 (0.03)** | **0.58 (0.06)** | **0.56 (0.06)** | 0.64 (0.03) | 0.72 (0.14) |
| Movie | GXI | 0.15 (0.13) | 0.34 (0.10) | 0.51 (0.10) | 0.42 (0.06) | 0.52 (0.10) | 0.61 (0.10) | 0.61 (0.09) | 0.51 (0.07) | 0.68 (0.12) | 0.76 (0.17) |
| | IG | **0.60 (0.06)** | 0.28 (0.12) | **0.69 (0.08)** | 0.00 (0.01) | **0.58 (0.21)** | **0.17 (0.06)** | **0.49 (0.09)** | **0.36 (0.11)** | 0.71 (0.03) | **0.56 (0.16)** |
| SciFact | GXI | 0.09 (0.19) | 0.27 (0.07) | 0.42 (0.15) | 0.34 (0.20) | 0.57 (0.20) | 0.55 (0.11) | 0.57 (0.09) | 0.59 (0.09) | 0.56 (0.11) | 0.63 (0.16) |
| | IG | **0.58 (0.14)** | 0.27 (0.09) | **0.46 (0.14)** | 0.01 (0.02) | **0.65 (0.19)** | **0.18 (0.19)** | **0.46 (0.08)** | **0.56 (0.11)** | 0.69 (0.03) | **0.54 (0.14)** |
| Twitter | GXI | 0.40 (0.09) | 0.41 (0.07) | 0.48 (0.10) | 0.48 (0.07) | 0.73 (0.10) | 0.51 (0.07) | 0.64 (0.05) | 0.75 (0.05) | 0.73 (0.06) | 0.64 (0.04) |
| | IG | **0.53 (0.06)** | 0.41 (0.08) | **0.70 (0.07)** | 0.40 (0.05) | 0.67 (0.13) | **0.36 (0.06)** | **0.62 (0.05)** | **0.59 (0.05)** | **0.71 (0.09)** | 0.66 (0.07) |

Table 6: Performance when we cumulatively add modifications to GradientXInput

| | | Comprehensiveness ↑ | | | | | Sufficiency ↓ | | | | |
| | | Gemma | LLAMA | | Qwen | | Gemma | LLAMA | | Qwen | |
| | | 2B | 1B | 3B | 1.5B | 3B | 2B | 1B | 3B | 1.5B | 3B |
| BoolQ | Gradient X Input | 0.09 (0.08) | 0.18 (0.07) | 0.45 (0.07) | 0.27 (0.13) | 0.57 (0.18) | 0.60 (0.09) | 0.71 (0.07) | 0.39 (0.07) | 0.54 (0.17) | 0.43 (0.11) |
| | + LN freeze | 0.08 (0.08) | 0.50 (0.07) | 0.45 (0.14) | 0.32 (0.26) | 0.43 (0.19) | 0.65 (0.10) | 0.36 (0.10) | 0.44 (0.16) | 0.55 (0.14) | 0.55 (0.15) |
| | + Grad norm | 0.44 (0.08) | 0.38 (0.08) | 0.48 (0.09) | 0.40 (0.16) | 0.63 (0.04) | 0.37 (0.09) | 0.56 (0.08) | 0.43 (0.08) | 0.52 (0.09) | 0.43 (0.07) |
| | + TSG | 0.09 (0.09) | 0.35 (0.11) | 0.25 (0.11) | 0.32 (0.19) | 0.47 (0.15) | 0.54 (0.11) | 0.56 (0.11) | 0.65 (0.09) | 0.72 (0.09) | 0.46 (0.10) |
| | + Grad norm + TSG | 0.37 (0.14) | 0.36 (0.09) | 0.49 (0.14) | 0.45 (0.07) | 0.67 (0.07) | 0.46 (0.11) | 0.57 (0.07) | 0.58 (0.04) | 0.68 (0.06) | 0.39 (0.10) |
| | + LN freeze + TSG | 0.09 (0.09) | 0.45 (0.07) | 0.47 (0.07) | 0.37 (0.29) | 0.18 (0.10) | 0.53 (0.11) | 0.24 (0.05) | 0.30 (0.12) | 0.62 (0.14) | 0.72 (0.19) |
| | + Grad norm + LN freeze | 0.53 (0.05) | 0.68 (0.03) | 0.72 (0.02) | 0.69 (0.07) | 0.66 (0.04) | 0.07 (0.05) | 0.28 (0.04) | 0.12 (0.04) | 0.12 (0.05) | 0.29 (0.10) |
| | + Grad norm + LN freeze + TSG | **0.59 (0.02)** | **0.69 (0.03)** | **0.72 (0.03)** | 0.68 (0.05) | 0.61 (0.03) | **0.03 (0.04)** | **0.22 (0.05)** | **0.10 (0.04)** | **0.09 (0.04)** | **0.23 (0.10)** |
| FEVER | Gradient X Input | 0.03 (0.19) | 0.39 (0.07) | 0.53 (0.12) | 0.43 (0.21) | 0.47 (0.13) | 0.47 (0.11) | 0.56 (0.07) | 0.63 (0.09) | 0.59 (0.06) | 0.73 (0.10) |
| | + LN freeze | 0.18 (0.17) | 0.49 (0.08) | 0.52 (0.09) | 0.37 (0.15) | 0.58 (0.14) | 0.38 (0.12) | 0.48 (0.08) | 0.69 (0.10) | 0.49 (0.09) | 0.61 (0.06) |
| | + Grad norm | 0.25 (0.12) | 0.40 (0.06) | 0.62 (0.08) | 0.53 (0.11) | **0.72 (0.09)** | 0.36 (0.10) | 0.54 (0.06) | 0.73 (0.05) | 0.51 (0.07) | 0.53 (0.07) |
| | + TSG | 0.03 (0.17) | 0.42 (0.09) | 0.41 (0.12) | 0.31 (0.07) | 0.47 (0.13) | 0.45 (0.11) | 0.54 (0.07) | 0.70 (0.07) | 0.70 (0.07) | 0.71 (0.09) |
| | + Grad norm + TSG | 0.30 (0.11) | 0.34 (0.08) | 0.61 (0.05) | 0.54 (0.06) | 0.70 (0.05) | 0.32 (0.12) | 0.55 (0.05) | 0.72 (0.05) | 0.71 (0.07) | 0.49 (0.08) |
| | + LN freeze + TSG | 0.10 (0.14) | 0.49 (0.08) | 0.38 (0.10) | 0.33 (0.17) | 0.43 (0.15) | 0.38 (0.13) | 0.38 (0.09) | 0.67 (0.09) | 0.65 (0.07) | 0.72 (0.12) |
| | + Grad norm + LN freeze | 0.39 (0.13) | 0.60 (0.03) | **0.76 (0.03)** | 0.62 (0.04) | 0.66 (0.04) | -0.03 (0.21) | 0.29 (0.07) | **0.39 (0.10)** | 0.25 (0.08) | 0.46 (0.08) |
| | + Grad norm + LN freeze + TSG | **0.42 (0.14)** | 0.62 (0.03) | 0.75 (0.02) | 0.51 (0.03) | 0.68 (0.04) | **-0.01 (0.19)** | **0.26 (0.07)** | 0.39 (0.11) | 0.31 (0.09) | **0.41 (0.08)** |
| HateXplain | Gradient X Input | 0.07 (0.05) | 0.53 (0.08) | 0.38 (0.14) | 0.49 (0.29) | 0.53 (0.15) | 0.59 (0.04) | 0.60 (0.06) | 0.67 (0.07) | 0.63 (0.09) | 0.63 (0.10) |
| | + LN freeze | 0.19 (0.07) | 0.42 (0.17) | 0.31 (0.18) | 0.19 (0.08) | 0.18 (0.10) | 0.58 (0.04) | 0.66 (0.06) | 0.77 (0.08) | 0.68 (0.05) | 0.78 (0.06) |
| | + Grad norm | 0.39 (0.03) | 0.40 (0.09) | 0.61 (0.04) | 0.30 (0.05) | **0.83 (0.06)** | 0.58 (0.03) | 0.59 (0.09) | 0.65 (0.05) | 0.67 (0.07) | 0.61 (0.04) |
| | + TSG | 0.03 (0.02) | 0.39 (0.11) | 0.47 (0.09) | 0.41 (0.10) | 0.42 (0.13) | 0.67 (0.04) | 0.65 (0.06) | 0.62 (0.05) | 0.75 (0.09) | 0.68 (0.08) |
| | + Grad norm + TSG | 0.46 (0.02) | 0.61 (0.04) | 0.64 (0.03) | 0.54 (0.07) | 0.81 (0.05) | 0.51 (0.04) | 0.45 (0.06) | 0.58 (0.06) | 0.73 (0.06) | **0.45 (0.05)** |
| | + LN freeze + TSG | 0.22 (0.05) | 0.57 (0.13) | 0.28 (0.12) | 0.17 (0.10) | 0.15 (0.08) | 0.56 (0.06) | 0.47 (0.09) | 0.70 (0.05) | 0.76 (0.06) | 0.80 (0.07) |
| | + Grad norm + LN freeze | 0.63 (0.02) | 0.67 (0.04) | 0.64 (0.06) | **0.85 (0.09)** | 0.50 (0.05) | 0.31 (0.05) | 0.37 (0.06) | 0.52 (0.05) | **0.28 (0.05)** | 0.91 (0.11) |
| | + Grad norm + LN freeze + TSG | **0.64 (0.01)** | **0.68 (0.04)** | 0.64 (0.08) | 0.80 (0.06) | 0.63 (0.08) | **0.26 (0.04)** | 0.26 (0.06) | 0.46 (0.04) | 0.35 (0.06) | 0.51 (0.15) |
| Movie | Gradient X Input | 0.15 (0.13) | 0.34 (0.10) | 0.51 (0.10) | 0.42 (0.06) | 0.52 (0.10) | 0.61 (0.10) | 0.61 (0.09) | 0.51 (0.07) | 0.68 (0.12) | 0.76 (0.17) |
| | + LN freeze | 0.28 (0.15) | 0.57 (0.06) | 0.68 (0.07) | 0.29 (0.10) | 0.26 (0.17) | 0.56 (0.11) | 0.28 (0.07) | 0.40 (0.10) | 0.75 (0.14) | 0.75 (0.08) |
| | + Grad norm | 0.52 (0.07) | 0.34 (0.08) | 0.49 (0.04) | 0.42 (0.13) | 0.73 (0.06) | 0.50 (0.06) | 0.65 (0.07) | 0.71 (0.05) | 0.55 (0.08) | 0.51 (0.07) |
| | + TSG | 0.14 (0.11) | 0.57 (0.08) | 0.27 (0.05) | 0.56 (0.11) | 0.50 (0.10) | 0.65 (0.10) | 0.45 (0.09) | 0.74 (0.05) | 0.68 (0.18) | 0.80 (0.18) |
| | + Grad norm + TSG | 0.59 (0.07) | 0.63 (0.05) | 0.49 (0.05) | 0.61 (0.07) | **0.88 (0.06)** | 0.49 (0.05) | 0.45 (0.05) | 0.67 (0.06) | 0.75 (0.11) | 0.43 (0.10) |
| | + LN freeze + TSG | 0.27 (0.14) | 0.62 (0.04) | 0.62 (0.08) | 0.37 (0.12) | 0.35 (0.28) | 0.56 (0.09) | 0.25 (0.05) | 0.35 (0.07) | 1.10 (0.19) | 0.76 (0.09) |
| | + Grad norm + LN freeze | 0.55 (0.04) | 0.68 (0.03) | **0.78 (0.02)** | 0.82 (0.04) | 0.73 (0.12) | 0.16 (0.06) | 0.26 (0.06) | 0.31 (0.05) | **0.13 (0.06)** | 0.45 (0.19) |
| | + Grad norm + LN freeze + TSG | **0.60 (0.03)** | **0.71 (0.04)** | 0.78 (0.02) | **0.83 (0.03)** | 0.76 (0.09) | **0.11 (0.06)** | **0.19 (0.06)** | **0.30 (0.05)** | 0.17 (0.05) | **0.31 (0.19)** |
| SciFact | Gradient X Input | 0.09 (0.19) | 0.27 (0.07) | 0.42 (0.15) | 0.34 (0.20) | 0.57 (0.20) | 0.55 (0.11) | 0.57 (0.09) | 0.59 (0.09) | 0.56 (0.11) | 0.63 (0.16) |
| | + LN freeze | 0.16 (0.17) | 0.52 (0.06) | 0.49 (0.13) | 0.23 (0.14) | 0.59 (0.28) | 0.45 (0.15) | 0.42 (0.09) | 0.55 (0.10) | 0.57 (0.11) | 0.65 (0.13) |
| | + Grad norm | 0.41 (0.12) | 0.36 (0.07) | 0.48 (0.11) | 0.43 (0.15) | **0.76 (0.09)** | 0.43 (0.16) | 0.53 (0.07) | 0.69 (0.10) | 0.55 (0.14) | 0.52 (0.09) |
| | + TSG | 0.09 (0.19) | 0.31 (0.11) | 0.25 (0.11) | 0.37 (0.14) | 0.63 (0.21) | 0.56 (0.12) | 0.56 (0.09) | 0.69 (0.09) | 0.80 (0.20) | 0.66 (0.13) |
| | + Grad norm + TSG | 0.43 (0.09) | 0.33 (0.09) | 0.51 (0.08) | 0.54 (0.07) | 0.74 (0.10) | 0.43 (0.14) | 0.53 (0.07) | 0.68 (0.07) | 0.75 (0.10) | 0.42 (0.12) |
| | + LN freeze + TSG | 0.23 (0.13) | 0.53 (0.07) | 0.34 (0.11) | 0.28 (0.11) | 0.36 (0.27) | 0.48 (0.12) | 0.33 (0.07) | 0.59 (0.08) | 0.74 (0.08) | 0.76 (0.15) |
| | + Grad norm + LN freeze | **0.60 (0.09)** | 0.64 (0.03) | 0.72 (0.04) | **0.66 (0.09)** | 0.72 (0.07) | 0.05 (0.18) | 0.25 (0.07) | **0.30 (0.09)** | **0.20 (0.08)** | 0.40 (0.12) |
| | + Grad norm + LN freeze + TSG | 0.57 (0.08) | **0.67 (0.03)** | **0.74 (0.04)** | 0.62 (0.08) | 0.74 (0.06) | **0.04 (0.18)** | **0.22 (0.08)** | 0.31 (0.09) | 0.26 (0.07) | **0.38 (0.11)** |
| Twitter | Gradient X Input | 0.40 (0.09) | 0.41 (0.07) | 0.48 (0.10) | 0.48 (0.07) | 0.73 (0.10) | 0.51 (0.07) | 0.64 (0.05) | 0.75 (0.05) | 0.73 (0.06) | 0.64 (0.04) |
| | + LN freeze | 0.45 (0.08) | 0.50 (0.06) | 0.46 (0.08) | 0.39 (0.05) | 0.75 (0.10) | 0.48 (0.05) | 0.56 (0.04) | 0.75 (0.06) | 0.72 (0.06) | 0.69 (0.05) |
| | + Grad norm | 0.48 (0.07) | 0.38 (0.09) | 0.55 (0.05) | 0.39 (0.08) | 0.69 (0.06) | 0.47 (0.04) | 0.62 (0.05) | 0.75 (0.02) | 0.69 (0.03) | 0.64 (0.04) |
| | + TSG | 0.35 (0.06) | 0.50 (0.06) | 0.47 (0.09) | 0.59 (0.06) | 0.71 (0.08) | 0.59 (0.06) | 0.61 (0.05) | 0.76 (0.05) | 0.78 (0.04) | 0.66 (0.05) |
| | + Grad norm + TSG | 0.47 (0.05) | 0.48 (0.05) | 0.55 (0.04) | 0.63 (0.06) | **0.80 (0.05)** | 0.48 (0.04) | 0.51 (0.04) | 0.68 (0.02) | 0.75 (0.04) | 0.61 (0.04) |
| | + LN freeze + TSG | 0.37 (0.07) | 0.47 (0.06) | 0.52 (0.09) | 0.42 (0.06) | 0.51 (0.14) | 0.56 (0.05) | 0.63 (0.05) | 0.75 (0.04) | 0.87 (0.05) | 0.73 (0.05) |
| | + Grad norm + LN freeze | **0.52 (0.06)** | 0.64 (0.03) | **0.73 (0.03)** | 0.78 (0.06) | 0.65 (0.05) | 0.32 (0.04) | 0.43 (0.04) | **0.60 (0.03)** | 0.47 (0.04) | 0.66 (0.08) |
| | + Grad norm + LN freeze + TSG | **0.52 (0.06)** | **0.68 (0.03)** | 0.72 (0.03) | 0.76 (0.06) | 0.72 (0.06) | 0.34 (0.04) | **0.41 (0.04)** | 0.61 (0.05) | 0.55 (0.04) | **0.58 (0.04)** |

**HateXplain** comprises Twitter posts, where the task is to classify whether a post contains hate speech Mathew et al. (2021)

We downsample our datasets because computing and evaluating feature attribution and circuit identification methods is computationally expensive. We create smaller subsets by randomly sampling 300 examples where all the models predicted the correct answer. We present an overview of the datasets in Table 7

Table 7: Dataset statistics

| Name | Number of examples | AVG number of tokens | Task |
|------|--------------------|-----------------------|------|
| Hatexplain | 300 | 314 | Hatespeech detection |
| Movie Review | 199 | 930 | Sentiment classification |
| Twitter Sentiment | 300 | 78 | Sentiment classification |
| BoolQ | 199 | 1184 | Question-answering |
| FEVER | 300 | 336 | Fact-verification |
| Scifact | 209 | 448 | Fact-verification |

## A.4 LICENSES

Here is a list of the assets used in our experiments and their Licences:

1. BoolQ: CC-BY 3.0

2. FEVER: CC-BY 3.0

3. SciFact: CC BY 4.0.

4. HateXplain: MIT License

5. Movie review: CC-BY 3.0

6. Twitter Sentiment Extraction: CC-BY 4.0

7. LLAMA models: LLAMA3.2 COMMUNITY LICENSE AGREEMENT

8. Gemma models: GEMMA LICENCE AGREEMENT

9. Qwen models: Qwen LICENSE AGREEMENT

## A.5 COMPUTATIONAL REQUIREMENTS

We ran all our experiments on an A100 80GB GPU. The feature attribution and self-repair experiments were fast, using less than an hour each for the smaller models (<3B) and around 2–3 hours for the large models. The experiments comparing circuit identification were slow, especially for integrated gradients. Integrated gradients used around half a day per dataset on the A100 GPU.

## A.6 CLASSIFICATION PERFORMANCE

In Table 8, we evaluate the large language models' performance on the datasets. All models perform well across datasets except for LLAMA-3.2 1B, which always predicts positive sentiment on the Movie and Twitter datasets (high recall and low specificity). Since we in the main paper focus on evaluating explanation faithfulness, a strong performance is not crucial for accurate results. We only want accurate explanations of the models' inner mechanisms, whether that inner mechanism solves the tasks well or not.

## A.7 MODEL PROMPTS

We designed unique prompt templates for each model-dataset pair. We tested different templates and chose the ones that resulted in the highest classification accuracy. Each prompt was designed such that the model would answer yes or no. The prompt templates are provided in our released code repository: https://anonymous.4open.science/r/explainable_transformer-D693

## A.8 TSG AT DIFFERENT TEMPERATURES

In Table 9, we present the GIM results for different temperatures used in TSG. Note that we ran this experiment after having selected the temperature of 2 in the main paper. We selected this temperature based on the results for Gemma 2B on Fever and HateXplain. Each model-dataset pair has a sweetspot. Increasing the temperature to high levels degrades performance.

Table 8: Classification performance of the models across the datasets. Except for LLAMA3.1 1B on the sentiment classification datasets, where it always predicts positive sentiment, the models perform well across all tasks.

|  |  | Accuracy | Precision | Recall | F1 | Specificity |
|---|---|---|---|---|---|---|
| HateXplain | Gemma2.2 2B | 57.49 | 50.39 | 98.82 | 66.74 | 26.09 |
|  | Gemma2.2 9B | 65.12 | 55.31 | 100.00 | 71.22 | 38.62 |
|  | Llama3.2 1B | 50.51 | 46.14 | 87.54 | 60.43 | 22.38 |
|  | Llama3.2 3B | 63.74 | 54.65 | 93.94 | 69.10 | 40.79 |
|  | Llama3.1 8B | 54.94 | 48.93 | 100.00 | 65.71 | 20.72 |
|  | Qwen2 1.5B | 54.36 | 48.55 | 95.79 | 64.44 | 22.89 |
|  | Qwen2 3B | 69.62 | 59.50 | 92.76 | 72.50 | 52.05 |
| Movie | Gemma2.2 2B | 82.41 | 100.00 | 64.65 | 78.53 | 100.00 |
|  | Gemma2.2 9B | 95.48 | 98.91 | 91.92 | 95.29 | 99.00 |
|  | Llama3.2 1B | 49.75 | 49.75 | 100.00 | 66.44 | 0.00 |
|  | Llama3.2 3B | 92.46 | 87.50 | 98.99 | 92.89 | 86.00 |
|  | Llama3.1 8B | 88.44 | 81.67 | 98.99 | 89.50 | 78.00 |
|  | Qwen2 1.5B | 85.93 | 80.34 | 94.95 | 87.04 | 77.00 |
|  | Qwen2 3B | 85.93 | 97.33 | 73.74 | 83.91 | 98.00 |
| Twitter | Gemma2.2 2B | 84.28 | 89.69 | 79.11 | 84.07 | 89.98 |
|  | Gemma2.2 9B | 88.11 | 89.37 | 87.77 | 88.56 | 88.48 |
|  | Llama3.2 1B | 52.69 | 52.58 | 99.83 | 68.88 | 0.71 |
|  | Llama3.2 3B | 84.06 | 91.34 | 76.91 | 83.50 | 91.95 |
|  | Llama3.1 8B | 87.06 | 85.43 | 90.81 | 88.04 | 82.92 |
|  | Qwen2 1.5B | 77.79 | 73.92 | 89.06 | 80.79 | 65.35 |
|  | Qwen2 3B | 81.82 | 93.15 | 70.53 | 80.28 | 94.28 |
| BoolQ | Gemma2.2 2B | 78.39 | 81.82 | 85.04 | 83.40 | 66.67 |
|  | Gemma2.2 9B | 82.41 | 87.10 | 85.04 | 86.06 | 77.78 |
|  | Llama3.2 1B | 65.33 | 69.86 | 80.31 | 74.73 | 38.89 |
|  | Llama3.2 3B | 67.84 | 87.95 | 57.48 | 69.52 | 86.11 |
|  | Llama3.1 8B | 76.88 | 89.32 | 72.44 | 80.00 | 84.72 |
|  | Qwen2 1.5B | 66.33 | 68.29 | 88.19 | 76.98 | 27.78 |
|  | Qwen2 3B | 64.32 | 87.84 | 51.18 | 64.68 | 87.50 |
| FEVER | Gemma2.2 2B | 81.41 | 85.19 | 75.70 | 80.17 | 87.04 |
|  | Gemma2.2 9B | 93.41 | 91.59 | 95.48 | 93.49 | 91.36 |
|  | Llama3.2 1B | 62.22 | 61.22 | 65.12 | 63.11 | 59.36 |
|  | Llama3.2 3B | 92.31 | 94.08 | 90.17 | 92.09 | 94.41 |
|  | Llama3.1 8B | 94.37 | 94.98 | 93.60 | 94.29 | 95.13 |
|  | Qwen2 1.5B | 92.10 | 94.09 | 89.71 | 91.85 | 94.44 |
|  | Qwen2 3B | 89.30 | 96.92 | 81.01 | 88.25 | 97.47 |
| SciFact | Gemma2.2 2B | 78.95 | 80.13 | 90.58 | 85.03 | 56.34 |
|  | Gemma2.2 9B | 86.60 | 92.31 | 86.96 | 89.55 | 85.92 |
|  | Llama3.2 1B | 49.76 | 77.97 | 33.33 | 46.70 | 81.69 |
|  | Llama3.2 3B | 82.78 | 91.13 | 81.88 | 86.26 | 84.51 |
|  | Llama3.1 8B | 83.73 | 94.83 | 79.71 | 86.61 | 91.55 |
|  | Qwen2 1.5B | 79.43 | 86.82 | 81.16 | 83.90 | 76.06 |
|  | Qwen2 3B | 77.03 | 97.87 | 66.67 | 79.31 | 97.18 |

Table 9: Faithfulness of GIM when using different temperatures for TSG across all models and datasets.

| | | Comprehensiveness ↑ | | | | | Sufficiency ↓ | | | | |
| | | Gemma | LLAMA | | Qwen | | Gemma | LLAMA | | Qwen | |
| | Temp | 2B | 1B | 3B | 1.5B | 3B | 2B | 1B | 3B | 1.5B | 3B |
|---|---|---|---|---|---|---|---|---|---|---|---|
| BoolQ | 1 | 0.53 | 0.68 | 0.72 | 0.69 | **0.66** | 0.07 | 0.28 | 0.12 | 0.12 | 0.29 |
| | 1.5 | 0.57 | 0.68 | 0.72 | 0.71 | 0.64 | 0.05 | 0.23 | 0.10 | 0.17 | **0.22** |
| | 2 | 0.59 | 0.69 | **0.72** | 0.68 | 0.61 | **0.03** | 0.22 | **0.10** | **0.09** | 0.23 |
| | 2.5 | 0.57 | 0.68 | 0.72 | 0.71 | 0.64 | 0.05 | 0.23 | 0.10 | 0.17 | **0.22** |
| | 3 | 0.60 | **0.69** | 0.70 | 0.67 | 0.61 | 0.04 | 0.21 | 0.12 | 0.10 | 0.25 |
| | 5 | 0.60 | 0.69 | 0.66 | **0.73** | 0.59 | 0.05 | **0.20** | 0.13 | 0.23 | 0.24 |
| | 10 | 0.60 | 0.69 | 0.65 | 0.62 | 0.62 | 0.06 | 0.20 | 0.12 | 0.22 | 0.23 |
| | 100 | **0.61** | 0.69 | 0.65 | 0.66 | 0.63 | 0.06 | 0.21 | 0.13 | 0.22 | 0.26 |
| FEVER | 1 | 0.39 | 0.60 | **0.76** | 0.62 | 0.66 | -0.03 | 0.29 | 0.39 | **0.25** | 0.46 |
| | 1.5 | 0.39 | 0.61 | 0.75 | **0.64** | 0.67 | 0.03 | **0.26** | **0.38** | 0.35 | 0.43 |
| | 2 | 0.42 | **0.62** | 0.75 | 0.51 | **0.68** | -0.01 | 0.26 | 0.39 | 0.31 | **0.41** |
| | 2.5 | 0.39 | 0.61 | 0.75 | **0.64** | 0.67 | 0.03 | **0.26** | **0.38** | 0.35 | 0.43 |
| | 3 | **0.43** | 0.62 | 0.74 | 0.48 | 0.63 | **-0.13** | 0.27 | 0.40 | 0.33 | 0.42 |
| | 5 | 0.42 | 0.61 | 0.74 | 0.55 | 0.63 | -0.13 | 0.27 | 0.41 | 0.48 | 0.44 |
| | 10 | 0.42 | 0.60 | 0.75 | 0.57 | 0.65 | -0.13 | 0.27 | 0.41 | 0.48 | 0.46 |
| | 100 | 0.42 | 0.60 | 0.75 | 0.57 | 0.67 | -0.12 | 0.27 | 0.41 | 0.47 | 0.48 |
| HateXplain | 1 | 0.63 | 0.67 | 0.64 | 0.85 | 0.50 | 0.31 | 0.37 | 0.52 | **0.28** | 0.91 |
| | 1.5 | **0.64** | 0.68 | **0.65** | 0.83 | 0.62 | **0.23** | 0.28 | 0.48 | 0.43 | 0.59 |
| | 2 | 0.64 | 0.68 | 0.64 | 0.80 | 0.63 | 0.26 | 0.26 | **0.46** | 0.35 | 0.51 |
| | 2.5 | **0.64** | 0.68 | **0.65** | 0.83 | 0.62 | **0.23** | 0.28 | 0.48 | 0.43 | 0.59 |
| | 3 | 0.61 | **0.68** | 0.59 | 0.77 | 0.56 | 0.29 | 0.22 | 0.49 | 0.36 | 0.46 |
| | 5 | 0.60 | 0.68 | 0.54 | 0.82 | 0.58 | 0.31 | **0.20** | 0.51 | 0.48 | 0.45 |
| | 10 | 0.61 | 0.68 | 0.56 | 0.77 | 0.68 | 0.33 | 0.21 | 0.52 | 0.52 | 0.45 |
| | 100 | 0.61 | 0.68 | 0.58 | **0.93** | **0.76** | 0.33 | 0.24 | 0.52 | 0.51 | **0.45** |
| Movie | 1 | 0.55 | 0.68 | 0.78 | 0.82 | 0.73 | 0.16 | 0.26 | 0.31 | **0.13** | 0.45 |
| | 1.5 | 0.58 | 0.70 | 0.78 | **0.84** | 0.75 | 0.12 | **0.18** | **0.28** | 0.23 | 0.38 |
| | 2 | 0.60 | 0.71 | **0.78** | 0.83 | **0.76** | **0.11** | 0.19 | 0.30 | 0.17 | 0.31 |
| | 2.5 | 0.58 | 0.70 | 0.78 | **0.84** | 0.75 | 0.12 | **0.18** | **0.28** | 0.23 | 0.38 |
| | 3 | 0.60 | 0.71 | 0.78 | 0.80 | 0.71 | 0.11 | 0.20 | 0.30 | 0.16 | 0.32 |
| | 5 | 0.60 | 0.71 | 0.76 | 0.82 | 0.72 | 0.11 | 0.21 | 0.32 | 0.28 | **0.31** |
| | 10 | 0.60 | **0.71** | 0.75 | 0.74 | 0.73 | 0.11 | 0.22 | 0.33 | 0.28 | 0.32 |
| | 100 | **0.61** | 0.71 | 0.75 | 0.74 | 0.70 | 0.14 | 0.23 | 0.34 | 0.28 | 0.42 |
| SciFact | 1 | **0.60** | 0.64 | 0.72 | 0.66 | 0.72 | 0.05 | 0.25 | 0.30 | **0.20** | 0.40 |
| | 1.5 | 0.58 | 0.66 | 0.73 | **0.70** | 0.73 | 0.06 | **0.22** | **0.30** | 0.34 | **0.38** |
| | 2 | 0.57 | **0.67** | **0.74** | 0.62 | **0.74** | 0.04 | 0.22 | 0.31 | 0.26 | 0.38 |
| | 2.5 | 0.58 | 0.66 | 0.73 | **0.70** | 0.73 | 0.06 | **0.22** | **0.30** | 0.34 | **0.38** |
| | 3 | 0.56 | 0.66 | 0.73 | 0.55 | 0.68 | **0.01** | 0.24 | 0.33 | 0.27 | 0.41 |
| | 5 | 0.54 | 0.64 | 0.72 | 0.61 | 0.67 | 0.01 | 0.24 | 0.35 | 0.46 | 0.41 |
| | 10 | 0.54 | 0.63 | 0.72 | 0.61 | 0.69 | 0.02 | 0.24 | 0.36 | 0.47 | 0.41 |
| | 100 | 0.55 | 0.63 | 0.73 | 0.63 | 0.70 | 0.04 | 0.24 | 0.36 | 0.46 | 0.43 |
| Twitter | 1 | 0.52 | 0.64 | 0.73 | 0.78 | 0.65 | **0.32** | 0.43 | **0.60** | **0.47** | 0.66 |
| | 1.5 | 0.52 | 0.66 | **0.73** | **0.81** | 0.71 | 0.33 | 0.42 | 0.61 | 0.58 | 0.59 |
| | 2 | 0.52 | 0.68 | 0.72 | 0.76 | **0.72** | 0.34 | **0.41** | 0.61 | 0.55 | 0.58 |
| | 2.5 | 0.52 | 0.66 | **0.73** | **0.81** | 0.71 | 0.33 | 0.42 | 0.61 | 0.58 | 0.59 |
| | 3 | 0.52 | **0.68** | 0.69 | 0.76 | 0.70 | 0.35 | 0.42 | 0.63 | 0.56 | 0.58 |
| | 5 | 0.52 | 0.67 | 0.69 | 0.79 | 0.69 | 0.35 | 0.43 | 0.64 | 0.60 | 0.58 |
| | 10 | 0.53 | 0.66 | 0.69 | 0.67 | 0.69 | 0.35 | 0.44 | 0.65 | 0.63 | **0.57** |
| | 100 | **0.54** | 0.65 | 0.69 | 0.67 | 0.72 | 0.35 | 0.45 | 0.65 | 0.63 | 0.58 |

## A.9 CIRCUIT IDENTIFICATION METHOD COMPARISON

In Figure 5, Figure 6, and Figure 7, we present the circuit identification method comparison for LLAMA-3.2 1B, Gemma-2 2B, and LLAMA-3.2 3B across all datasets. We see similar results as in the main paper: GIM substantially improves the faithfulness in the earliest layers. Computing Integrated Gradients per layer takes several days per dataset for LLAMA-3.2 3B and Gemma-2 2B on an A100 GPU.

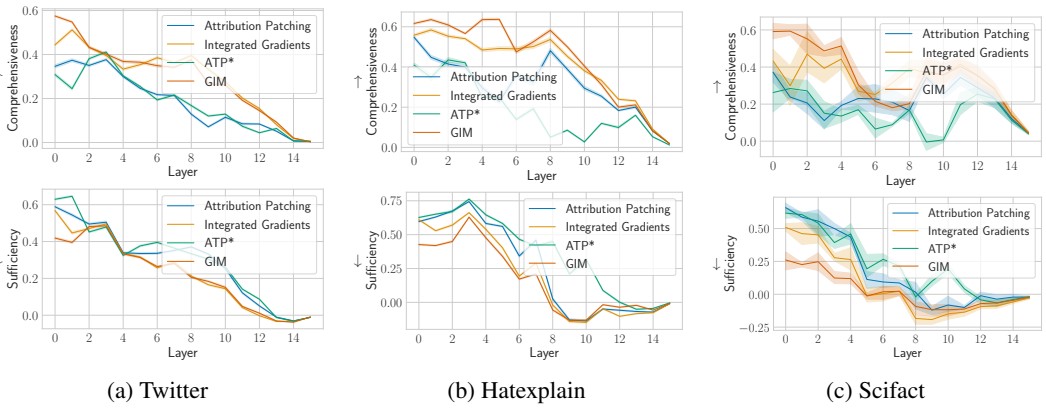

(a) Twitter          (b) Hatexplain          (c) Scifact

Figure 5: Faithfulness per layer for **LLAMA-3.2 1B**. The top row depicts comprehensiveness per layer, where higher is better. The bottom row depicts sufficiency, where lower is better.

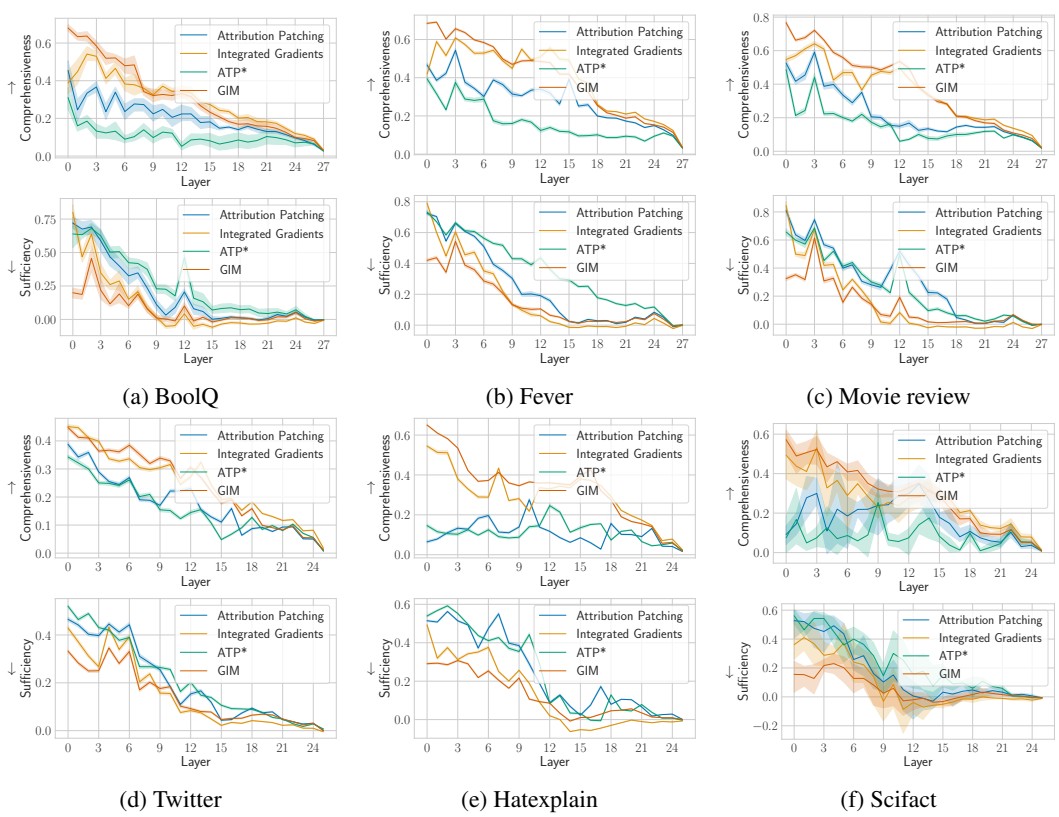

(a) BoolQ          (b) Fever          (c) Movie review

(d) Twitter          (e) Hatexplain          (f) Scifact

Figure 6: Faithfulness per layer for **Gemma-2 2B**. The top row depicts comprehensiveness per layer, where higher is better. The bottom row depicts sufficiency, where lower is better.

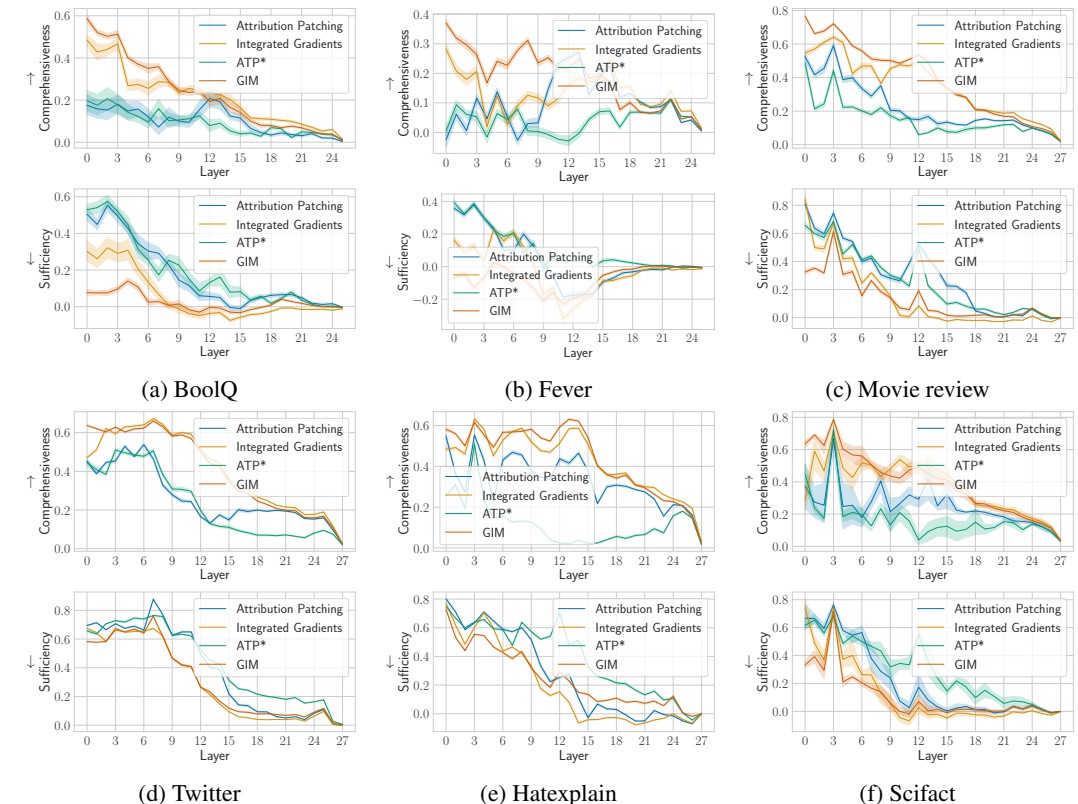

(a) BoolQ  (b) Fever  (c) Movie review

(d) Twitter  (e) Hatexplain  (f) Scifact

Figure 7: Faithfulness per layer for **LLAMA-3.2 3B**. The top row depicts comprehensiveness per layer, where higher is better. The bottom row depicts sufficiency, where lower is better.

## A.10 SELF-REPAIR RESULTS

In Figure 8, Figure 9, Figure 10, and Figure 11, we present additional empirical evidence for self-repair. We compare the joint ablation effect with the gradients, TSG, and the sum of individual ablation effects. The figures show similar results to those in the main paper. For models such as Qwen, the gradients are generally very large, which results in many points being plotted.

## A.11 SELF-REPAIR IN THREE ATTENTION SCORES

In the main paper, we only analyze self-repair for two attention scores; however, self-repair can occur for more attention scores. In Figure 13, Figure 14, Figure 15, and Figure 16, we present additional empirical evidence for self-repair. We compare the joint ablation effect with the gradients, TSG, and the sum of individual ablation effects. TSG better approximates the joint ablation effect than the other two.

## A.12 SELF-REPAIR RESULTS WITH HIGHER AND LOWER THRESHOLD

To detect self-repair in the main paper, we used a threshold of 0.1 on the coefficient of variation. This threshold was chosen empirically. In Figure 18, we show the self-repair results using different thresholds (0.05, 0.1, and 0.2) for LLAMA 3-1B on the Fever and HateXplain datasets. Increasing the threshold results in more points in the figures, where the additional points have less self-repair. Using alternative thresholds does not alter the paper's findings.

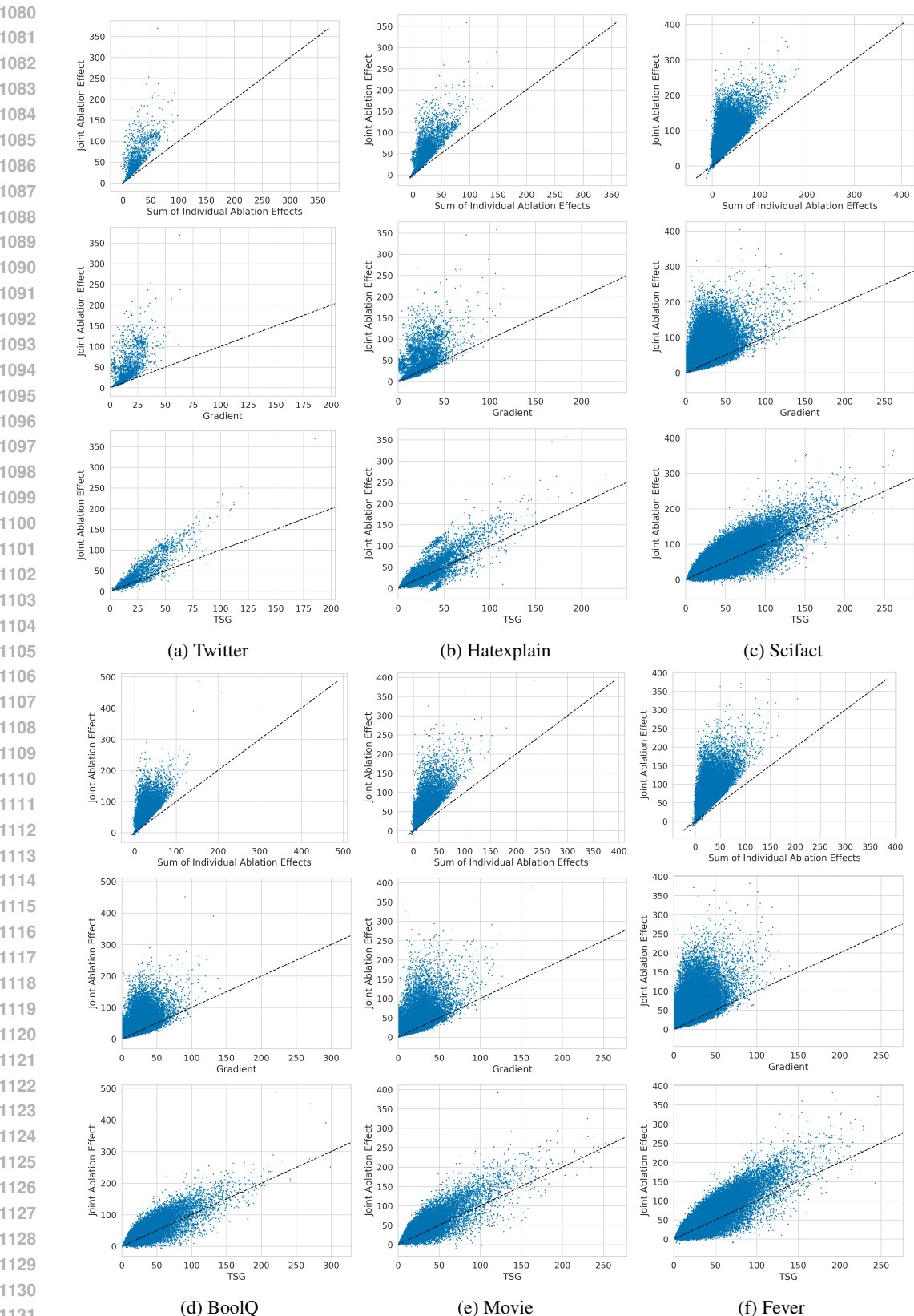

Figure 8: Self-repair for **Gemma-2 2B** and how TSG increases the attributions for the attention scores with the strongest self-repair effects

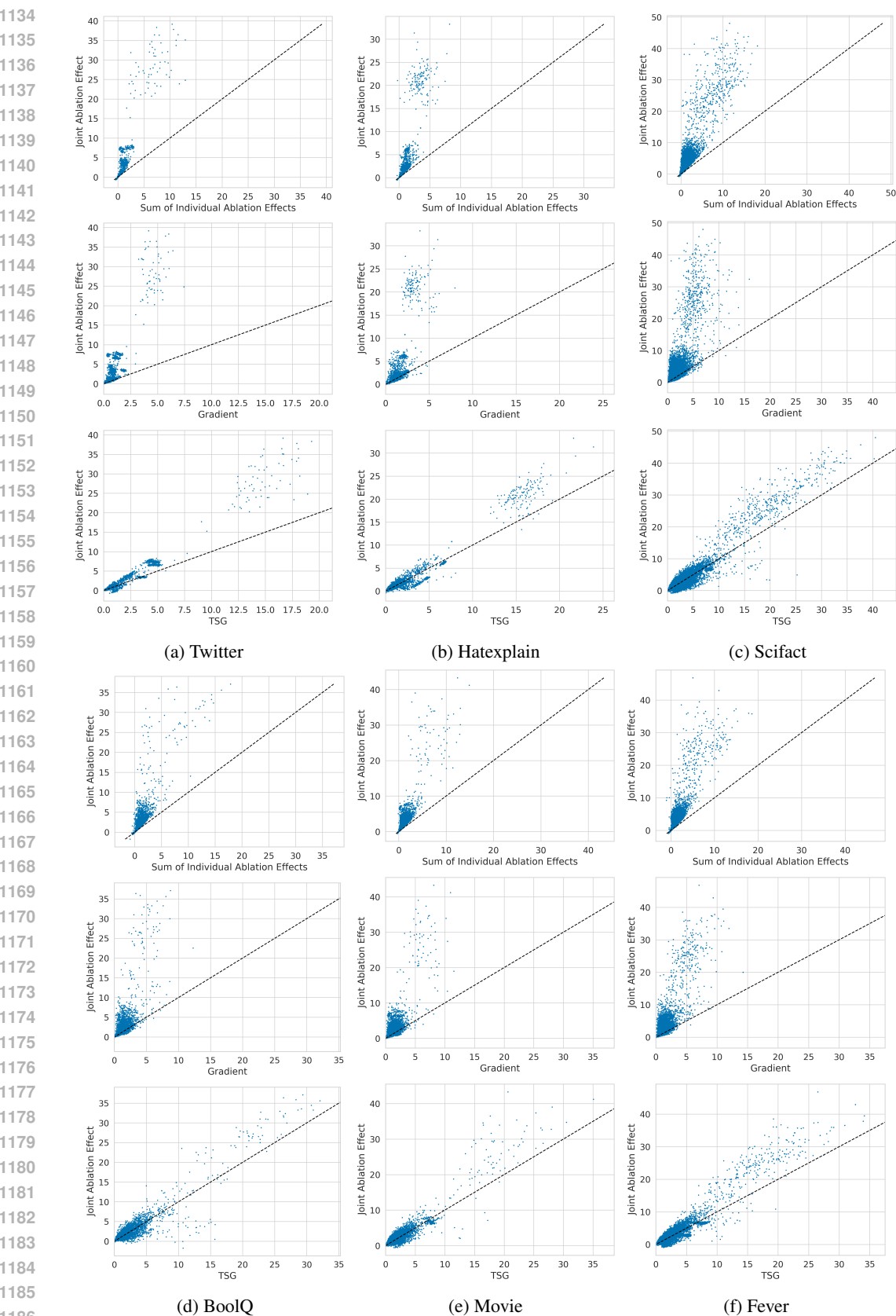

(a) Twitter   (b) Hatexplain   (c) Scifact

(d) BoolQ   (e) Movie   (f) Fever

Figure 9: Self-repair for **LLAMA-3.2 1B** and how TSG increases the attributions for the attention scores with the strongest self-repair effects

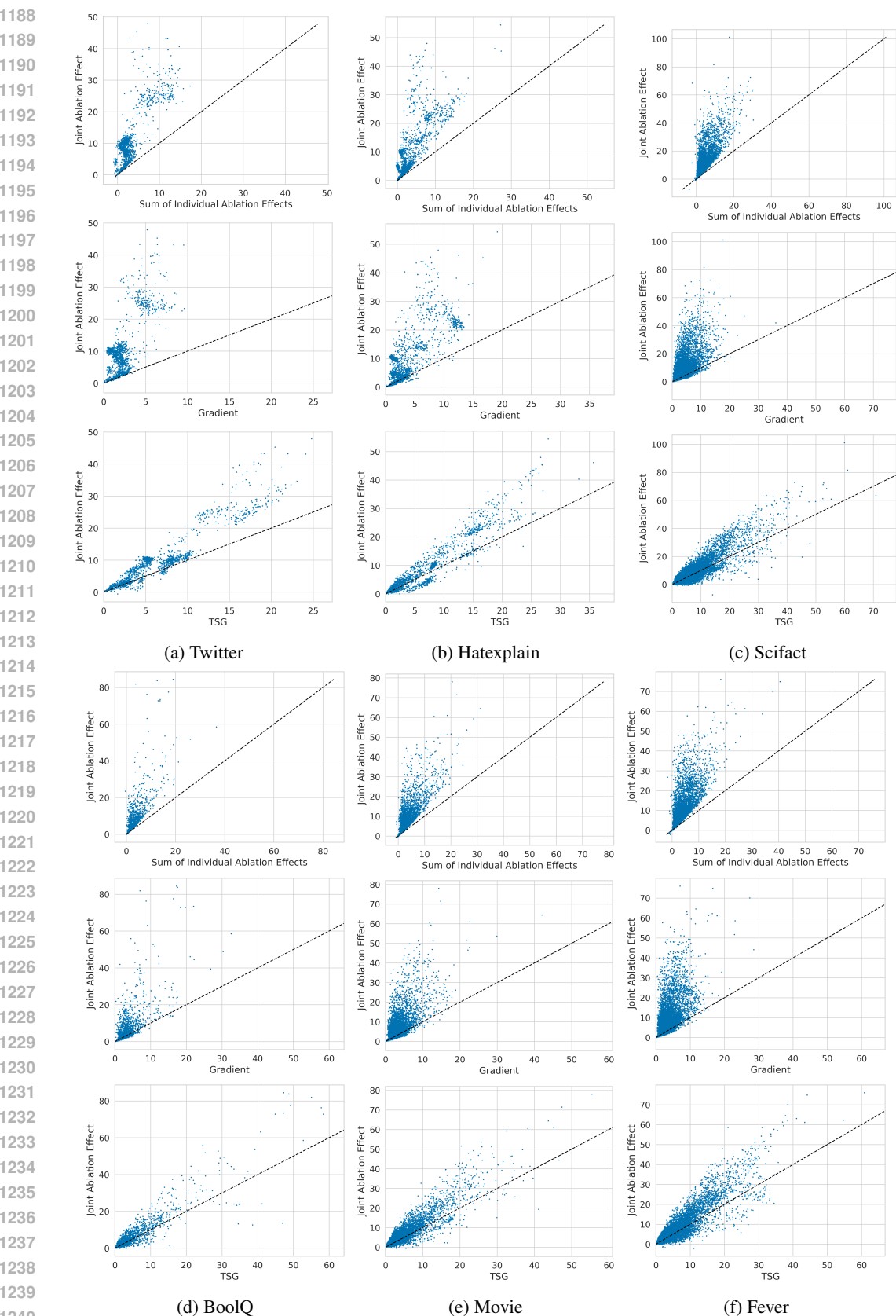

Figure 10: Self-repair for **LLAMA-3.2 3B** and how TSG increases the attributions for the attention scores with the strongest self-repair effects

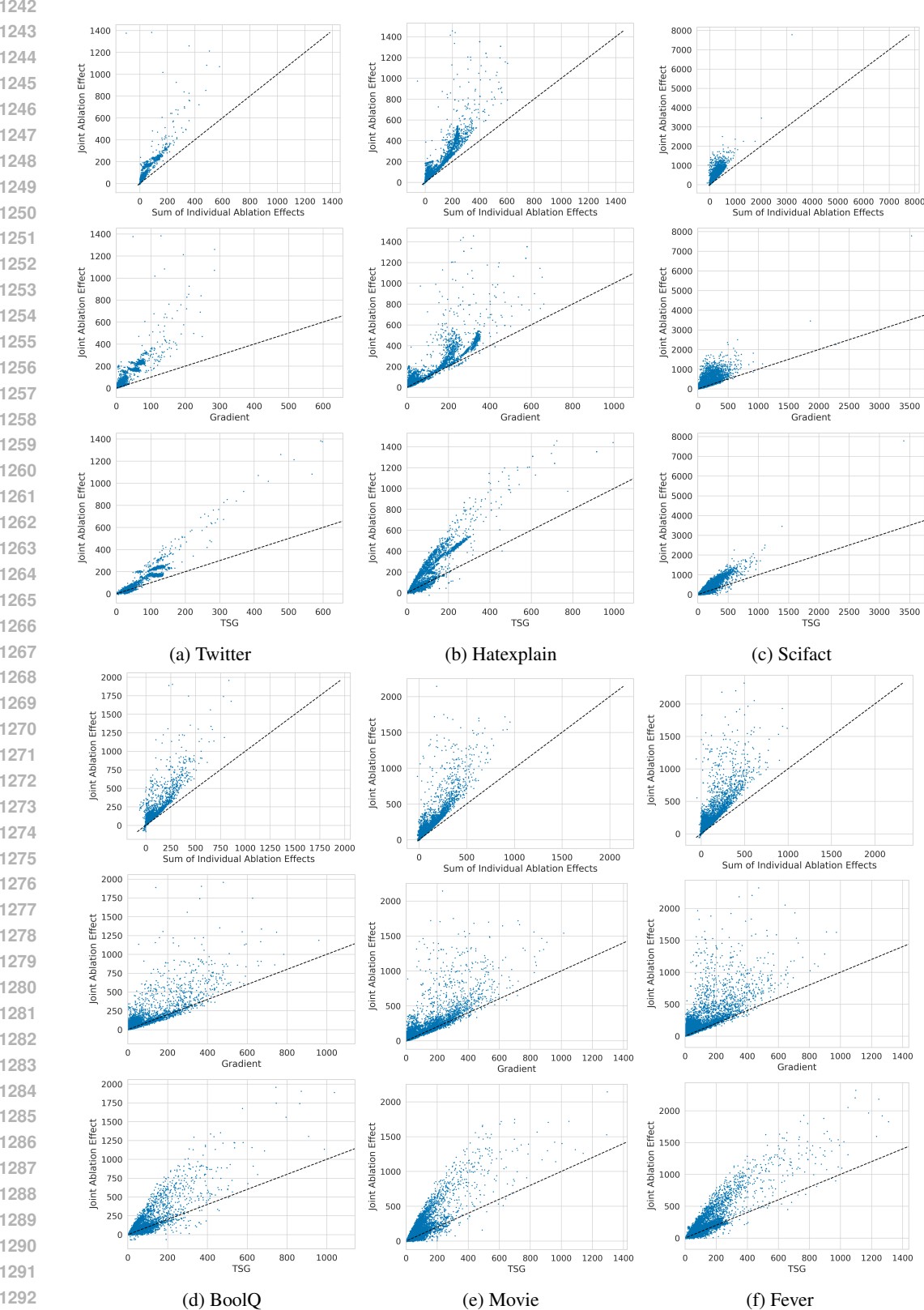

Figure 11: Self-repair for **Qwen-2.5 1.5B** and how TSG increases the attributions for the attention scores with the strongest self-repair effects

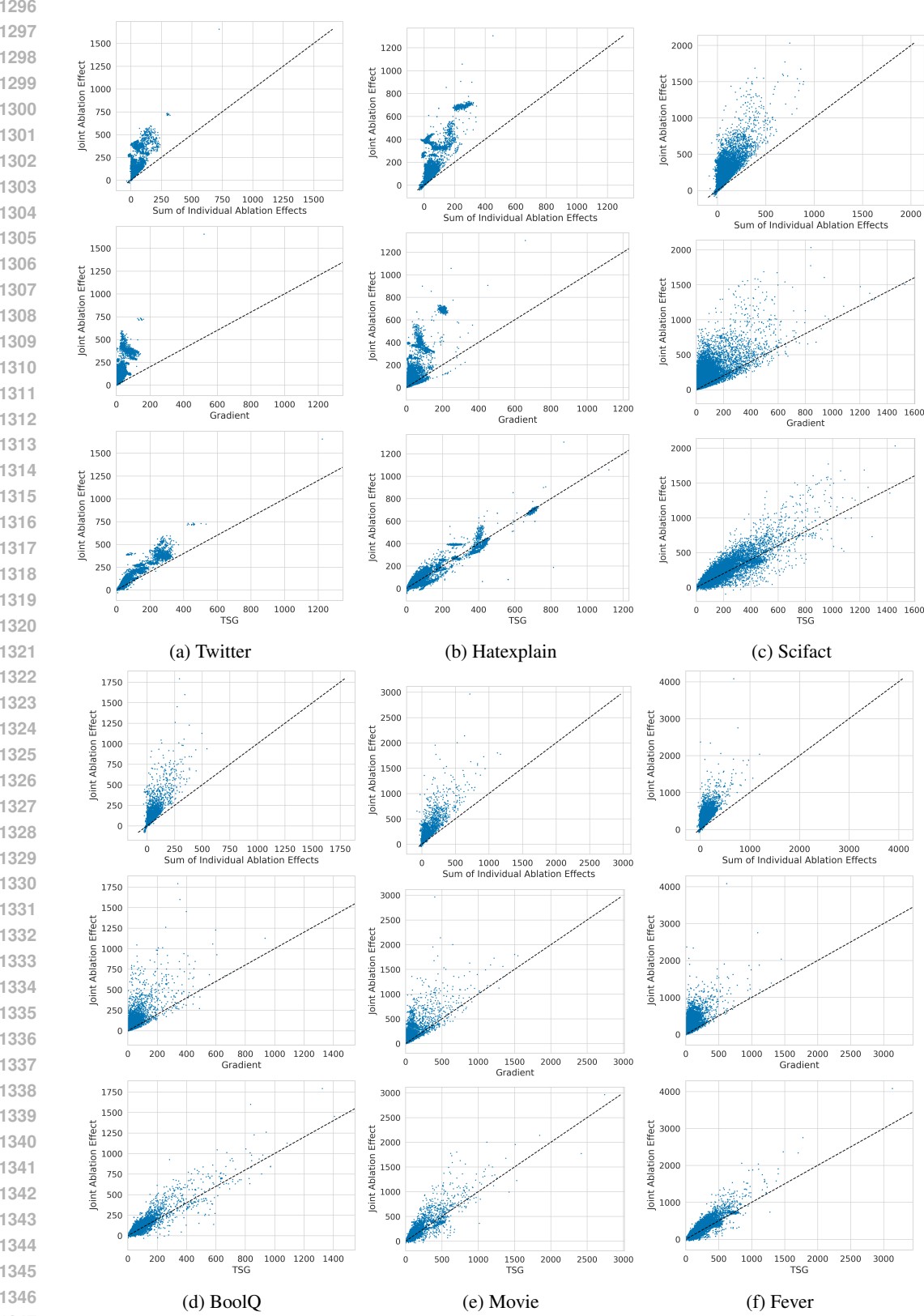

Figure 12: Self-repair for **Qwen-2.5 3B** and how TSG increases the attributions for the attention scores with the strongest self-repair effects

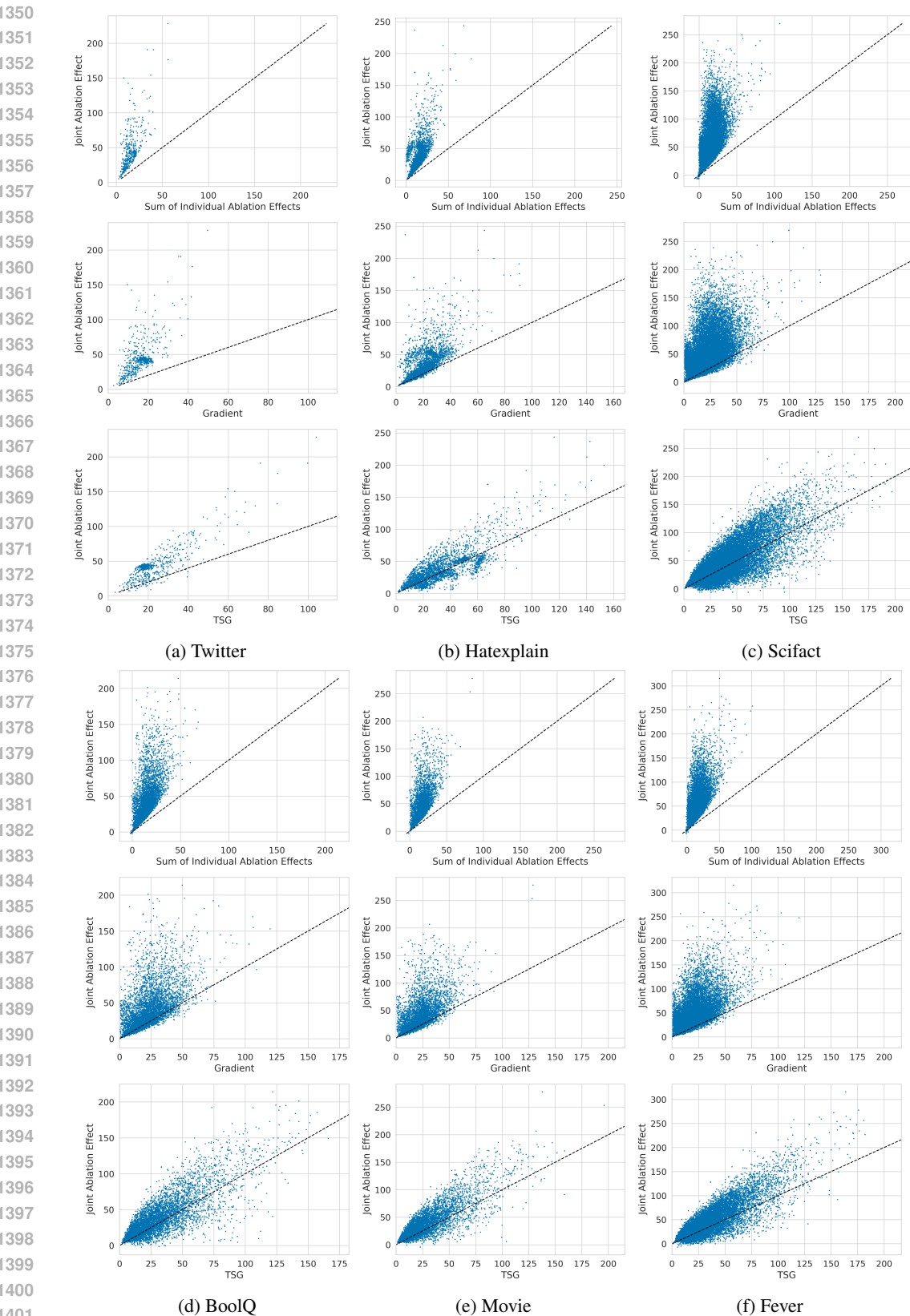

Figure 13: Self-repair in three attention scores for **Gemma-2 2B** and how TSG increases the attributions for the attention scores with the strongest self-repair effects

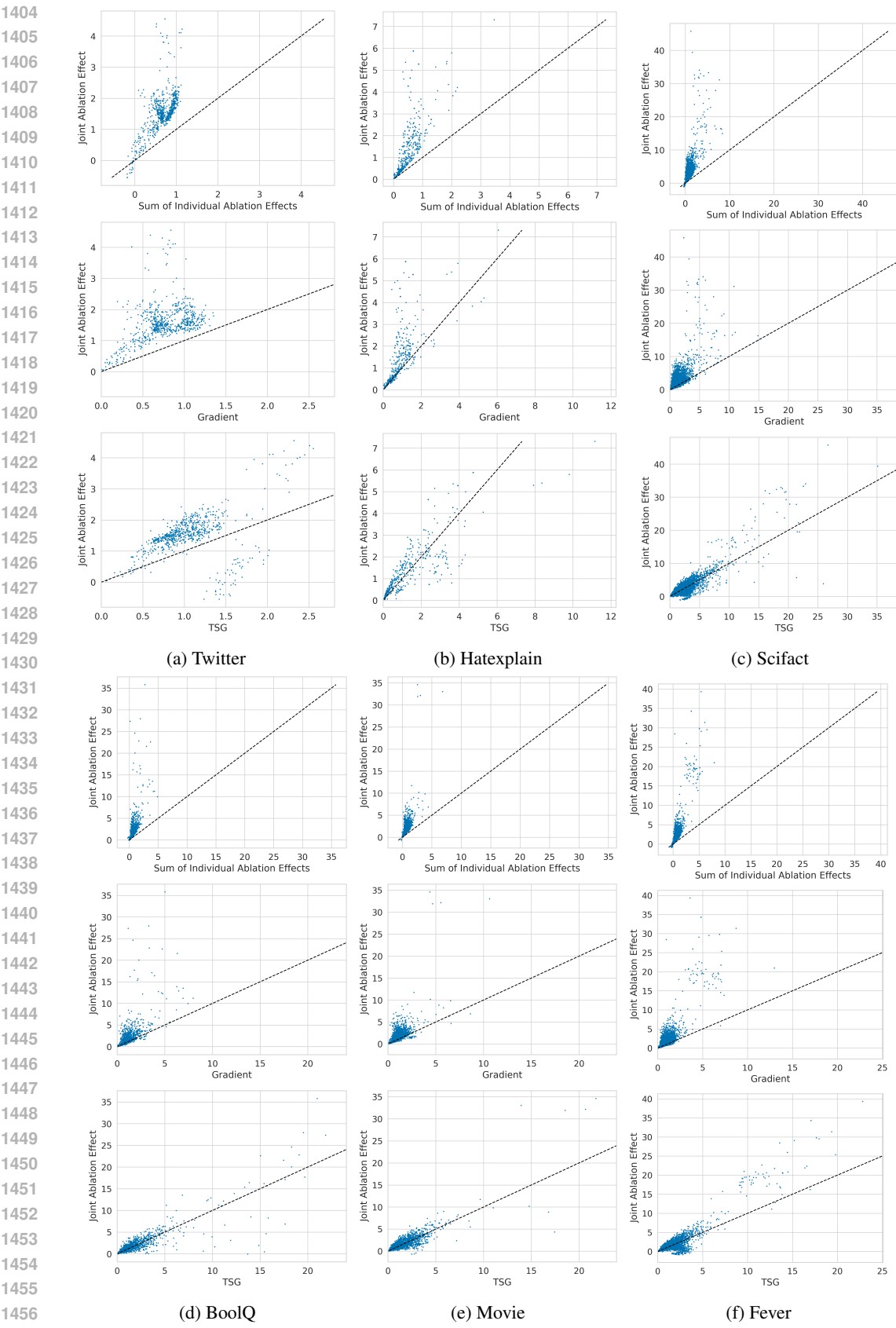

Figure 14: Self-repair in three attention scores for **LLAMA-3.2 1B** and how TSG increases the attributions for the attention scores with the strongest self-repair effects

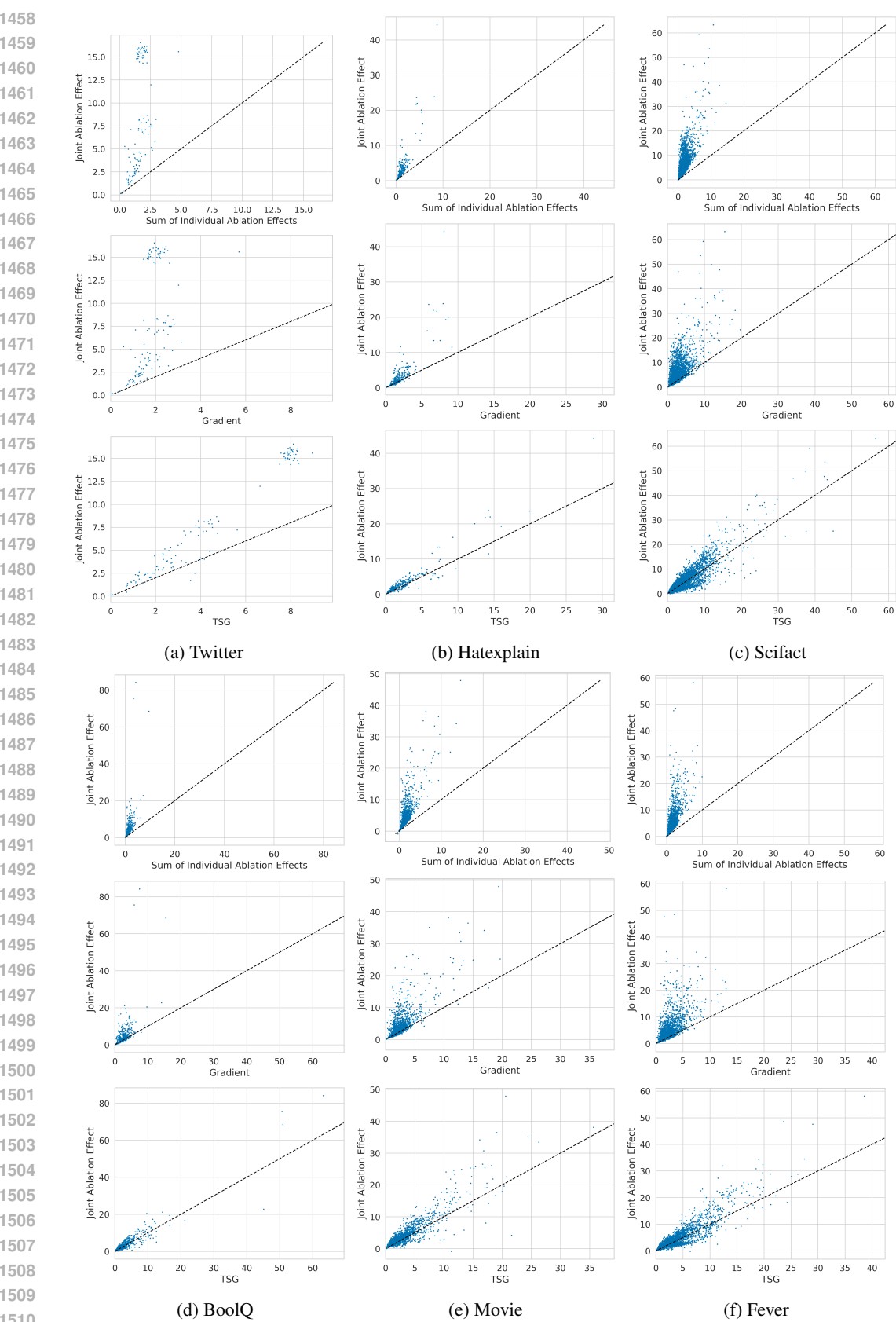

Figure 15: Self-repair in three attention scores for **LLAMA-3.2 3B** and how TSG increases the attributions for the attention scores with the strongest self-repair effects

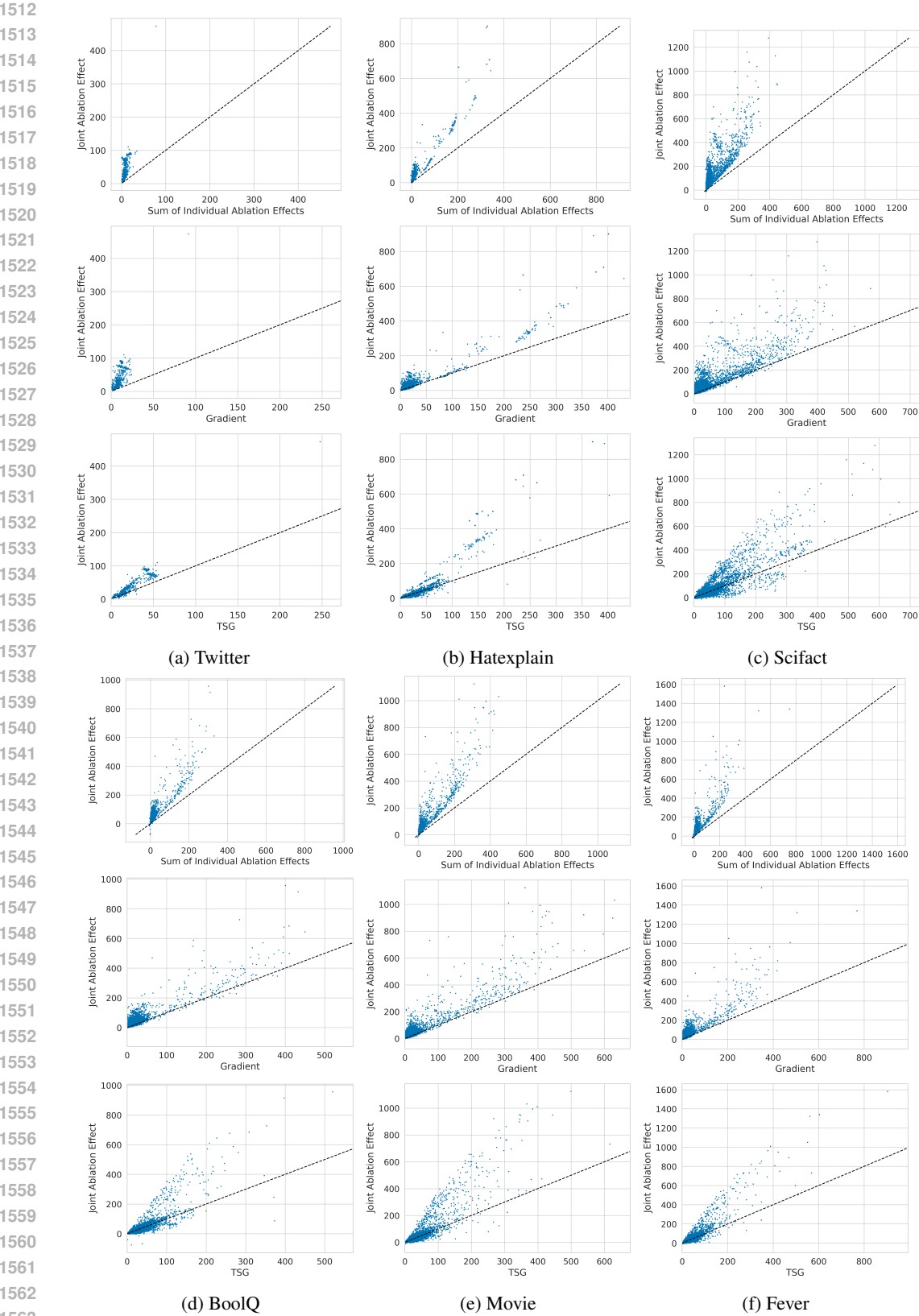

Figure 16: Self-repair in three attention scores for **Qwen-2.5 1.5B** and how TSG increases the attributions for the attention scores with the strongest self-repair effects

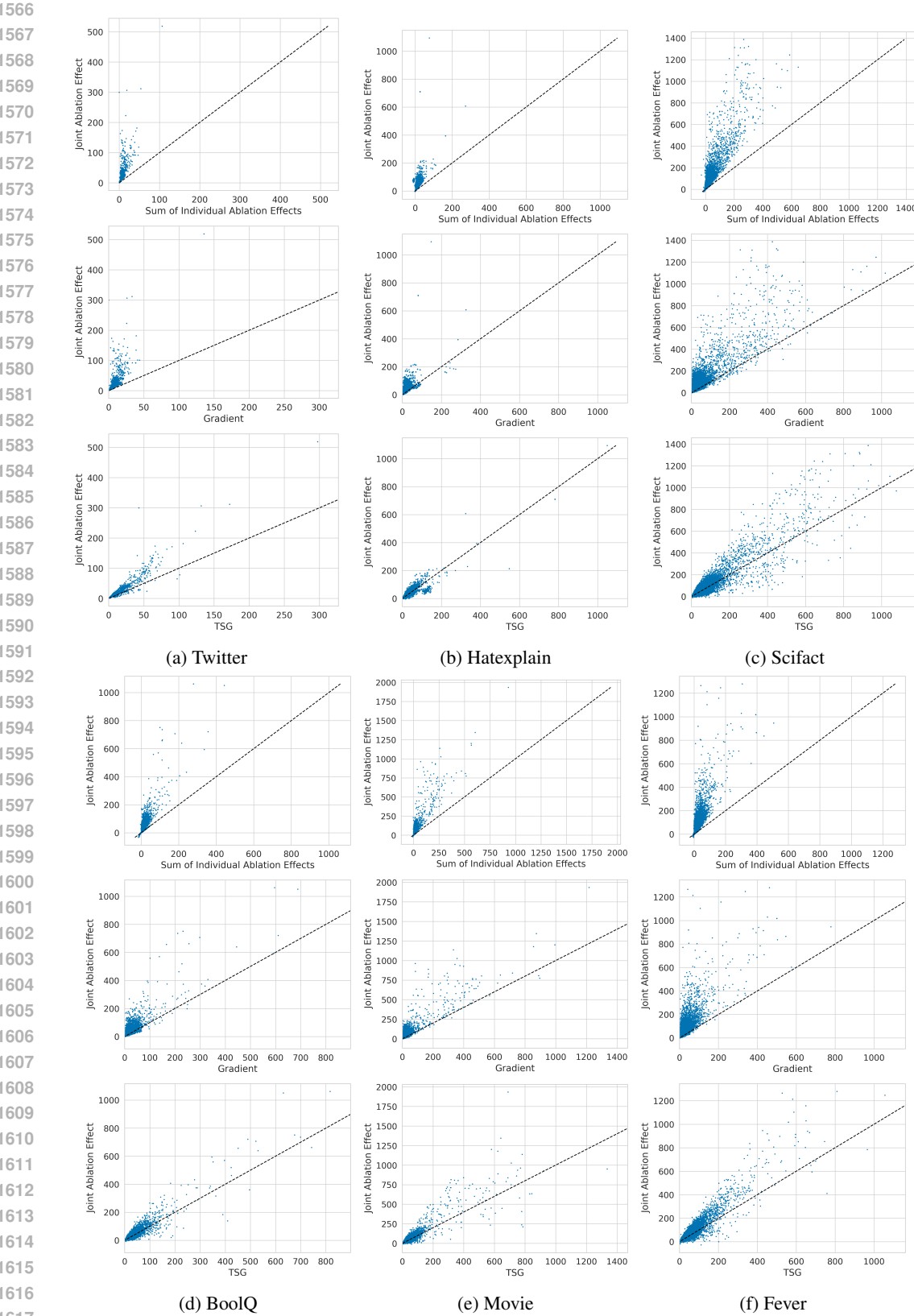

Figure 17: Self-repair in three attention scores for **Qwen-2.5 3B** and how TSG increases the attributions for the attention scores with the strongest self-repair effects

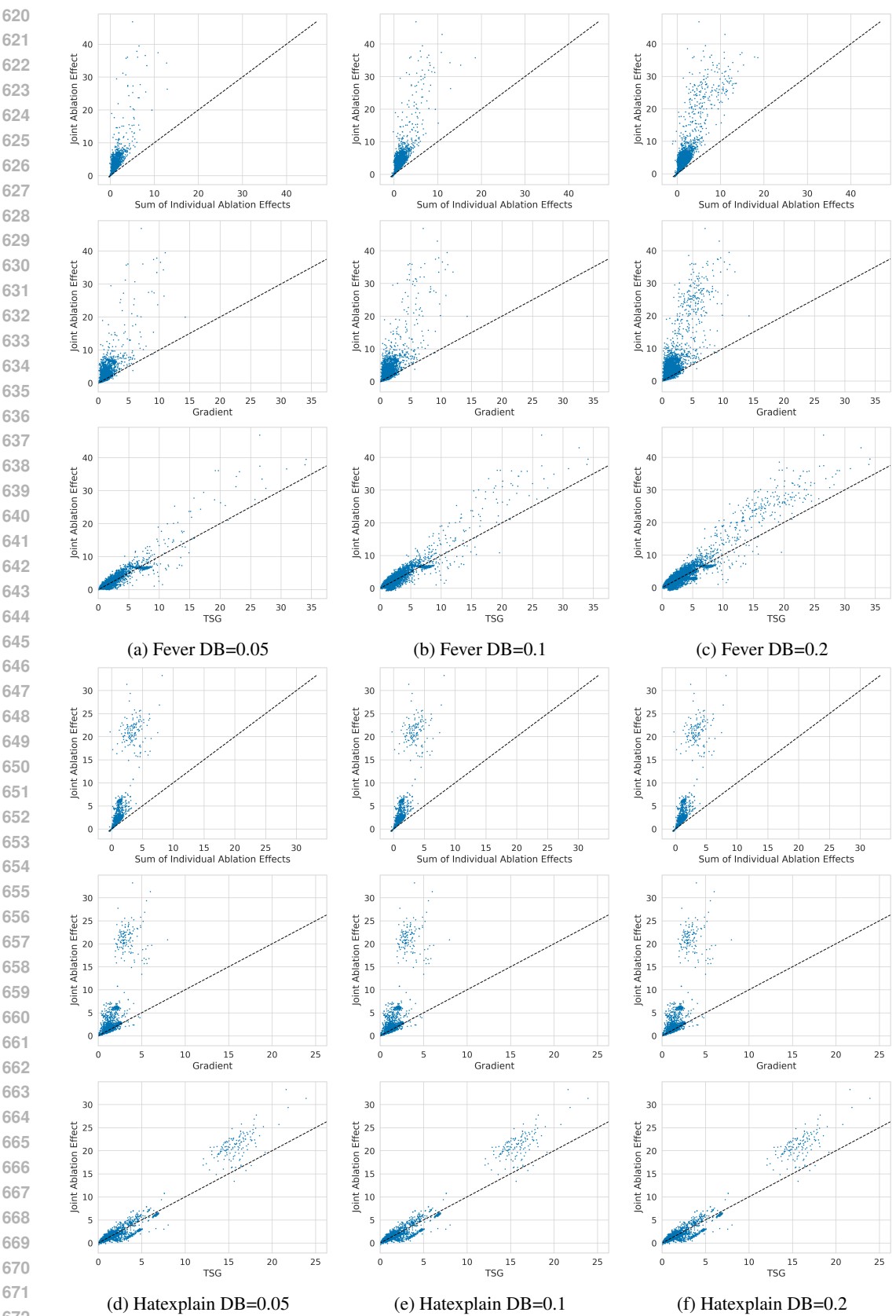

Figure 18: Self-repair for **LLAMA-3.2 1B** using different thresholds for the coefficient of variation. The threshold determines the number of points in the figures, where the additional points have less self-repair.

