# OpenReview forum: "GIM: Improved Interpretability for Large Language Models"
_ICLR.cc/2026/Conference — Submitted to ICLR 2026_

### Official Review · Reviewer_NupE · 2025-10-28

**Soundness:** 3
**Presentation:** 2
**Contribution:** 3
**Rating:** 4
**Confidence:** 3

**Summary:**

This paper identifies attention self-repair, where softmax redistribution within the attention mechanism masks the importance of individual scores, leading gradients to underestimate their true contribution.
To address this, the authors propose Gradient Interaction Modification (GIM), which adjusts the softmax gradient via temperature scaling to mitigate this effect.
The paper claims improved faithfulness over prior gradient-based feature attribution methods across several datasets and models.

**Strengths:**

1. **Novel diagnostic observation:** The paper identifies and analyzes the “attention self-repair” phenomenon—an overlooked cause of gradient underestimation in attention-based attribution methods. This diagnostic contribution is conceptually interesting and relevant to the interpretability community.

2. **Clear motivation and simple, insightful remedy:** The motivation around self-repair is well-grounded, and the proposed gradient modification (TSG) provides a simple yet insightful approach that could inspire follow-up research on faithful attention attribution.

3. **Lightweight and easily reproducible implementation:** Despite its conceptual simplicity, GIM can be applied as a small backward-pass modification without retraining, making it readily testable in other interpretability pipelines and future model analyses.

**Weaknesses:**

1. **Local-only correction:** GIM modifies gradients within each layer but ignores inter-layer coupling through residual and normalization paths. The method remains a local fix rather than a globally consistent gradient correction.

2. **Inter-head redundancy omitted:** Self-repair and TSG are defined per head; cross-head compensation and shared-output interactions are unaddressed.

3. **Top-2 simplification:** Figure 2, which forms a central motivation of the paper, measures self-repair using only the two largest attention scores, missing higher-order redundancy among 3–4 correlated tokens.

4. **Input-length sensitivity ignored:** No experiments are provided to analyze the effect of input token length on TSG behavior.

5. **Limited task scope:** Experiments are restricted to binary classification; the generality of the phenomenon to more complex tasks is unclear.

**Questions:**

**Questions for the Authors**

1. Have you analyzed self-repair effects that arise across multiple layers or between different attention heads? If such evidence exists, including those results would significantly strengthen the paper.

2. Why were only the two largest attention scores used in measuring self-repair (Figure 2)? Have you examined whether including top-3 or top-4 attention weights changes the measured frequency or strength of self-repair?

3. What is the rationale for selecting “significant attention weights” as > 0.01 (lines 305–306)? Was this empirically tuned, and how sensitive are your results to this threshold? If available, please include supporting results.

4. Table 6 shows minimal sensitivity to $T$. What explains this stability?

5. Since softmax normalization can be influenced by sequence length, did you analyze how TSG behaves on longer versus shorter inputs? Would an adaptive or length-dependent temperature yield more stable results?

6. Have you tested GIM on non-binary or more complex tasks (e.g., multi-class or open-ended settings)?

**Additional Suggestions**

1. Table 1 appears to present mean results only. Including standard deviations or confidence intervals would improve statistical transparency.

2. Clarify experimental setup: The main text (Eq. 8 and Eq. 9) does not specify layer-wise evaluation, yet Figure 3 reports layer-wise metrics. It would be helpful if the authors could explicitly clarify how layer-wise results were derived, perhaps in the appendix if space is limited.

3. Minor presentation issue: Table 3 appears truncated—please correct this formatting problem.

---

> ### Author Response · Authors · 2025-11-17
> **Revisions and response to weaknesses**
>
> Dear reviewer,
> Thank you for your detailed review. We have revised the paper based on your feedback and that of the other reviewer. Below, we address the weaknesses of the paper:
> 1. The attention self-repair is caused by the normalization effect in the softmax activation function. The softmax normalization effect does not occur between layers, so there is no inter-layer coupling through residual and normalization paths. Due to the chain rule, changes in backpropagation in one layer impact the entire model. The method is therefore a global fix, not local.
> 2. Again, softmax normalization does not occur between attention heads. There are no cross-head interactions for the attention self-repair that is caused by the softmax.
> 3. While our examples and experiments focus on pair-wise interactions, the theory of the paper applies to more than two attention scores. The math in 3.1 does not depend on a pair of competing attention scores but on a set. This set can have any number of attention scores. To also empirically demonstrate higher-order redundancy, we have added the results for self-repair for three attention scores in the revised paper (A.11 and Figures 13, 14, 15, and 16). The results are similar to the pair-wise interactions we showed in the paper, which supports our theory.
> 4. We agree that input length is important to consider. Conceptually, TSG depends on how many tokens receive non-negligible attention within each head rather than the sequence length. As the sequence length grows, if a head’s attention either remains local or stays uniform, then both the standard gradient and TSG are unchanged. However, if longer inputs introduce more competing positions, then we expect TSG to be more helpful than in the shorter context. In Table 7 (revised paper), we show the input lengths of each dataset. The lengths vary a lot. Out of our datasets, BoolQ has the longest average sequence length (1,184 tokens), and Table 4 shows that the TSG has a large positive effect when added. The Twitter dataset has the shortest average sequence length (78 tokens), and TSG is less beneficial than for the longer datasets.
> 5. This is true for the main paper, but in Tables 2 and 3 in the appendix, we evaluate the methods on multiple tasks that are not binary classification. Results are consistent whether it is binary classification or not.
>
>
> Next, we address your questions:
>
>
> 1. See our response to weaknesses 1 and 2.
> 2. See our response to weakness 3.
> 3. You are correct that our threshold of 0.1 is not based on principled grounding. We found the threshold empirically by testing lower and higher thresholds. We have added this to the appendix (A.12). In Figure 18, we present the results obtained using thresholds of 0.05, 0.1, and 0.2. Using different thresholds does not significantly change the results. A higher threshold adds points to the figure, where the additional points have less self-repair. Note also that this threshold is for illustrative purposes. Self-attention is a continuous phenomenon rather than a discrete one, and our method does not depend on a threshold.
> 4. Increasing T results in more uniform softmax weights. At a certain point, the softmax weights will be completely uniform, and increasing T further will not make a difference.
> 5. See our response to weakness 4.
> 6. See our response to weakness 5.
>
> **Response to suggestions:**
>
> 1. Good suggestion. We did not include uncertainties since we wanted the large table to be readable, but you and another reviewer have convinced us that we should include them. Therefore, in the revised paper, we show the standard deviations in Table 1.
> 2. Good point. We have added a clarified sentence after Equations 8 and 9 in the revised paper.
> 3. Thank you. We have addressed the issue in the revised paper.
>
> **Additional changes.**
>
> GIM substantially outperforms other circuit discovery methods on the mechanistic interpretability benchmark (A.1).  GIM being state-of-the-art on this benchmark is a significant contribution, but we added it to the appendix (A.1) rather than including it in the main paper. With the extra page in the camera-ready version, we will move these results from the appendix to the main paper and list them as one of our main contributions.

---

> > ### Comment · Reviewer_NupE · 2025-11-26
> >
> > Dear authors,
> >
> > Thank you for the detailed response. I appreciate the additional experiments and clarifications you have incorporated.
> >
> > That said, I still find the explanation regarding temperature scaling incomplete. As for the response to my Question 4, you mention that increasing $T$ eventually yields uniform softmax weights, and that after a certain point the weights will become completely uniform such that further increases in $T$ make no difference. If this were the primary mechanism, we would expect a monotonic trend or at least a clear pattern in the results as $T$ increases.
> >
> > However, the empirical results in Table 6 fluctuate irregularly across temperatures and show no sign of convergence or saturation. In addition, the differences across temperatures are extremely small—often within or below the standard deviations you report in the revised Table 1. This makes it difficult to attribute the performance changes to the mechanism described in your response, and raises questions about whether temperature scaling is producing a meaningful causal effect or simply introducing minor random variation.
> >
> > Additional clarification on this point would be appreciated.

---

> ### Author Response · Authors · 2025-11-27
>
> Dear Reviewer,
>
> Thank you for your detailed follow-up. We appreciate the opportunity to provide further clarification regarding the temperature scaling results.
>
> You are correct that the results in Table 9 do not show a uniform pattern of convergence across all experiments as $T$ increases. We agree that our previous explanation regarding "saturation" was imprecise. We would like to address your specific concerns about the fluctuations and provide evidence that the improvement offered by TSG is a meaningful causal effect rather than random variation.
>
> **1. Explaining the Fluctuations:**
> You correctly pointed out that if the mechanism were purely about reaching uniform weights, results should stabilize at high $T$. However, the goal of TSG is not uniformity, but rather to find the specific point where self-repair is broken (unmasking suppressed tokens) while preserving the signal of how the attention mechanism attended.
>
> * **The "Sweet Spot":** We achieve the best faithfulness results just as the temperature becomes high enough to break the self-repair within the attention head.
> * **The "Uniform Stability":** If $T$ increases far beyond this point, the softmax becomes effectively uniform. While the results do eventually stabilize here, they stabilize at a worse performance level because the uniform gradient destroys the information regarding which tokens were attended to.
>
> Between the sweet spot and the uniform stability, there will be random fluctuations in faithfulness as the softmax weights gradually become more uniform and the signal of the attention matrix is lost. The specific temperature required to break self-repair and the point where the softmax degrades into uniformity differ significantly across models and tasks due to varying magnitudes of the pre-softmax attention scores. For instance, tasks that require reasoning over a few tokens elicit sharper attention patterns than tasks that require reasoning over many tokens. Consequently, these cases require a higher temperature to break the winner-takes-all dynamic. Similarly, models like Qwen often generate very large attention scores compared to Gemma. For some model-task combinations, the attention scores are so large that the softmax distribution is not yet fully uniform even at the highest temperatures tested in Table 9, which explains why those specific results have not yet converged. The fluctuations of the sweet spot and the timing of convergence reflect these varying sensitivities rather than random noise.
>
> **2. Comparing faithfulness for different temperatures**
> Given this variability, we do not view specific high-temperature values (e.g., comparing $T=2$ vs. $T=2.5$) as the primary indicator of the method's validity. Instead, the critical evaluation of the causal effect lies in the comparison between the baseline ($T=1$) and the modified gradient ($T > 1$).
>
> The hypothesis is that the standard softmax gradient ($T=1$) mechanically masks self-repair. The "average best" heuristic of $T=2$ is intended to break this dynamic across the majority of cases.
>
> **3. Evidence of TSG not introducing random variation**
> To address your concern that TSG might simply be introducing random variation, we point to the asymmetry of our results in the Ablation Study (Table 4).
>
> * **Statistically Significant Upside:** Removing TSG causes statistically significant drops in faithfulness for several setups (e.g., BoolQ on Gemma 2B, Twitter on LLaMA 1B).
> * **Minimal Downside:** Importantly, TSG almost never substantially degrades performance. In the vast majority of cases, it either significantly improves the results or results in little difference.
>
> If TSG were merely adding random noise, we would expect to see performance degradation as often as improvement. Instead, we see a "safe" modification that aids or maintains faithfulness. The only exception is Qwen 0.5B, where TSG sometimes substantially worsens the results. However, while TSG does not improve every single case, this variation is standard for the field; as we show in Table 5, even established methods like Integrated Gradients do not outperform standard gradients across all model-dataset pairs. Similar to TSG, Integrated Gradients performs worse than standard gradients on Qwen 0.5B, yet Integrated Gradients remains a critical contribution to the field. The mostly positive (or neutral) impact of TSG, backed by statistical significance in the ablation study, confirms that the temperature adjustment produces a meaningful causal effect.

---

### Official Review · Reviewer_zyGM · 2025-10-30

**Soundness:** 3
**Presentation:** 3
**Contribution:** 2
**Rating:** 4
**Confidence:** 4

**Summary:**

The paper introduces an attribution method (GIM) for LLMs. Attribution methods are a key tool for interpretability research as they allow us to quickly identify causally important components of models or inputs. GIM is a combination of existing methods plus the newly-developed temperature-adjusted softmax gradient (TSG) method.

The paper formalizes the attention self-repair problem where if two positions provide redundant information, and both have high attention scores, ablating one of the two causes little change in the output (the attention weights focus on the other position and obtain the same information, leading to the same output).

TSG aims to address the attention self-repair issue by increasing the softmax temperature during the backward pass to yield gradients that provide better attributions.

GIM is compared to alternative attribution methods, and shown to perform the best when rated on comprehensiveness and sufficiency. Ablations to identify the impact of TSG and other individual ingredients of GIM are performed, showing that each ingredient adds to the performance of GIM.

Mandatory disclosure of LLM usage by the reviewer: LLMs were used to format the review text into prose and lists.

**Strengths:**

1. The paper identifies self-repair due to attention as a clear failure-mode of conventional gradient attributions and illustrates it with a toy example (Figure 1).

2. It shows that self-repair occurs in a variety of LLMs and tasks (Figure 2a and appendices).

3. The paper combines several existing tools (layernorm freeze, gradient normalization) and the novel TSG into a single method (GIM).

4. The metrics (sufficiency and comprehensiveness) seem well-chosen for this problem, directly benchmarking attribution scores by how well they predict cumulative ablations.

5. The paper thoroughly compares GIM to methods from the literature (GradientXInput, Integrated Gradients, DeepLIFT, TransformerLRP, AttnLRP), across multiple models and datasets.

6. The paper performs extensive ablations (Figure 4, Table 3) to test which improvements contribute to better attribution.

**Weaknesses:**

1. The paper describes TSG as an empirically-developed method that provides better attributions for OR-gate-like components. This seems well-supported by the mathematical and empirical arguments. However, I wonder whether this method is addressing a symptom or fixing the underlying issue of the attribution: Does TSG still work on non-OR-gate-like settings? Does it make attributions worse in some settings? This is somewhat discussed in the discussion section but left for future work; I would expect to see this investigation in the current paper as proposing TSG as a modification to attribution methods is the central claim of the paper.

2. The statistical significance of the improvement adding TSG is not convincing: Figure 4 compares the attributions with and without TSG. The boxplots are quite close & overlapping. It would be useful to quantify how significant the difference is; potentially also to analyse whether this improvement is on data points where attention self-repair was previously identified. Table 3 only lists point estimates without uncertainties.

**Questions:**

See weaknesses.

---

> ### Author Response · Authors · 2025-11-17
> **Revisions and response to weaknesses**
>
> Dear reviewer,
>
> Thank you for your thoughtful review. We have revised the paper based on your feedback and that of the other reviewer. Below, we address the weaknesses of the paper:
>
> 1. We acknowledge that deeper analysis would strengthen the paper. Theoretically, TSG still works on non-OR-gate examples and does generally not worsen attributions. Consider, for instance, the following 3 examples:
>    1. Attention is approximately uniform. In that case, softmax(x) is approximately softmax(x/T), and the gradient remains largely unchanged
>    2. Attention has multiple, approximately equal peaks. In that case, we encounter the self-attention problem described in Section 3, and the temperature scaling breaks the self-attention mechanism.
>    3. Attention has a single, strong peak. In that case, the single token will remain the dominant one.
>
>    In situations where self-repair is present, TSG fixes exactly the problem we identified. In situations where self-repair is not present, because Softmax is monotonic, TSG preserves the relative order between the logits; it only scales the difference. At most, it can collapse to a tie as T goes to infinity. However, we used T=2, which has a modest effect.
>
>
>    To show the above empirically, we conducted several experiments but ultimately excluded them due to inconclusive results:
>    1. We attempted to evaluate TSG on examples where attention self-repair occurs versus where it does not. However, self-repair occurs in every example, making this comparison infeasible.
>    2. We attempted to apply TSG only to attention scores where we detected self-repair during backpropagation; however, this worsened performance. The issue is that self-repair is continuous rather than binary. When using a threshold to detect self-repair, we amplify specific attention scores more than others, creating imbalances that result in some components receiving disproportionate attribution. This imbalance propagates through backpropagation to earlier layers.
>
>
> 2. The boxplots overlap because there is high inter-dataset variance. Within each model-dataset pair, the overlap is smaller. We have therefore updated the ablation study in the revised paper. There are now three ablation study tables in the appendix (Tables 4, 5, and 6). In Table 4, we show the impact of ablating a modification from GIM. We show the mean values and the standard deviations. The ablations that deteriorate the performance of GIM are shown in bold. Those that are significant are underlined (independent t-test $0.05<\alpha$). Across most model-data pairs, all three modifications are important. For some model-data pairs, ablating a modification will improve the performance. Variance is expected, and in Table 5, we demonstrate this. Integrated gradients are generally considered superior to GradientXInput in the field. In Table 5, we compare the two methods. While integrated gradients are generally better, the results vary across models and datasets. This shows that we cannot expect improvements to be consistent across datasets and models. Table 6 is what used to be Table 3, but with standard deviations (as you requested).
>
> You requested an analysis of TSG comparing examples where self-repair occurs with those where it doesn’t. As mentioned in weakness 1, self-repair occurs in every example, making it infeasible to investigate examples without it.
>
> **Additional changes.**
> GIM substantially outperforms other circuit discovery methods on the mechanistic interpretability benchmark (A.1).  GIM being state-of-the-art on this benchmark is a significant contribution, but we added it to the appendix (A.1) rather than including it in the main paper. With the extra page in the camera-ready version, we will move these results from the appendix to the main paper and list them as one of our main contributions.

---

> > ### Comment · Reviewer_zyGM · 2025-11-19
> >
> > Dear authors,
> >
> > Thank you for the detailed response.
> >
> > 1. Your examples illustrate well that these typical cases should not qualitatively be changed by TSG. To answer how the quantitative accuracy of attributions I turn to the second point.
> >
> > 2. I appreciate the very detailed statistical information added to the paper. The results (e.g. Table 6) don't convince me that TSG (now called TGM?) improves the performance over existing methods. For many columns, the standard deviation is larger than the difference between scores. Based on the t-test results, TSG is better in about half the cases. Thus I remain skeptical whether TSG actually helps or hurts performance.
> >
> > I remain skeptical, and believe the evidence presented here does not support the claim "that GIM significantly improves
> > faithfulness over existing circuit identification and feature attribution methods" in the abstract.
> >
> > I applaud the authors for providing a detailed analysis with extensive statistical detail though, and I think the paper should be accepted if its claims were adjusted to reflect the evidence presented.

---

> ### Author Response · Authors · 2025-11-19
> **Revised contribution statement**
>
> Thank you for your reply!
>
> TGM was an error. We have revised the paper and changed it to TSG.
>
> As requested, we have also revised the contributions statement in the paper from:
>
> "We empirically isolate and demonstrate the importance of each of our three modifications, showing their significant positive effect on explanation faithfulness across multiple large language models and tasks."
>
> To:
>
> "We empirically isolate and analyze the importance of each of our three modifications across multiple large language models and tasks."

---

> > ### Comment · Reviewer_zyGM · 2025-11-19
> >
> > Thanks for the quick response!
> >
> > Please make sure you edit this everywhere. When I open the paper (+ check on OR), the abstract still says
> >
> > > We introduce Gradient Interaction Modifications (GIM), a technique that accounts for self-repair during backpropagation. Extensive experiments [...] demonstrate that GIM significantly improves faithfulness over existing circuit identification and feature attribution methods.

---

> ### Author Response · Authors · 2025-11-19
>
> We might have misunderstood you. We thought the problem was the claim "TSG significantly improves the faithfulness," not "GIM (i.e., the combination of all three modifications) significantly improves faithfulness."
>
> Based on the results in Tables 1 and 2, and Figures 3, 5, 6, and 7, do you not think we have sufficient evidence to claim that GIM (the combination of all modifications) improves faithfulness compared to other feature attribution and circuit discovery methods?

---

> > ### Comment · Reviewer_zyGM · 2025-11-19
> >
> > The paper claims improvements "over existing circuit identification and feature attribution methods"; the other two components are existing methods.

---

> ### Author Response · Authors · 2025-11-19
>
> Good point.
>
> We changed it to:
>
> Extensive experiments across multiple large language models (Gemma 2B/9B, LLAMA 1B/3B/8B, Qwen 1.5B/3B) and diverse tasks demonstrate that GIM frequently achieves state-of-the-art results on circuit identification and feature attribution.

---

> > ### Author Response · Authors · 2025-11-19
> >
> > In the contribution statement, we also changed:
> >
> > We demonstrate that GIM is more faithful than other gradient-based circuit identification and feature attribution methods across multiple large language models (Gemma 2B/9B, LLAMA 1B/3B/8B, and Qwen 1.5B/3B) and six datasets spanning four tasks (question-answering, fact-verification, sentiment classification, and hate speech detection)
> >
> > To:
> >
> > We demonstrate that GIM is **often** more faithful than other gradient-based circuit identification and feature attribution methods across multiple large language models (Gemma 2B/9B, LLAMA 1B/3B/8B, and Qwen 1.5B/3B) and six datasets spanning four tasks (question-answering, fact-verification, sentiment classification, and hate speech detection)

---

### Official Review · Reviewer_hYi1 · 2025-10-31

**Soundness:** 2
**Presentation:** 3
**Contribution:** 3
**Rating:** 4
**Confidence:** 3

**Summary:**

This paper identifies a novel self-repair phenomenon within the attention mechanism of transformers. When values at positions with large attention weights contribute similarly toward the output, the effect of ablating an individual attention weights is obfuscated by softmax normalization, mirroring OR-gate like behavior. The authors demonstrate the phenomenon of attention self-repair on a variety of datasets and models by showing that the effect of jointly ablating two attention scores is much stronger than the sum of the effect of ablating the individual attention scores. To mitigate this in the context of attribution, the authors introduce gradient interaction modifications (GIM), a collection of three individual components (temperature-adjusted softmax gradients, layernorm freeze, and gradient normalization). Using GIM, they often achieve higher comprehensiveness and lower sufficiency when ablating model components deemed crucial to the output as compared to other attribution methods.

**Strengths:**

1. The mathematical explanation of the self repair mechanism for attention weights is clear and convincing.
2. The proposed TSG solution is well motivated theoretically.
3. The breadth of models and datasets is quite comprehensive.
4. GIM as a whole often outperforms prior attribution methods in comprehensiveness and sufficiency across a variety of models and datasets.

**Weaknesses:**

1. The depth of analysis for the attention self-repair effect is somewhat weak. While the same analysis is present across multiple datasets and models, it seems limited to comparing the joint and sum of individual ablation effects. Are there any other analyses that the authors investigated to better understand this novel phenomenon, such as the average group size of similarly contributing values with substantial attention weights, or how many groups of such values there tend to be?
2. Looking at Table 3 in the appendix (which is slightly cut off), it seems that the attribution performance improvement of TSG alone upon Gradient X Input is relatively minor and weaker than either of the two previously established techniques it is combined with, LN freeze and grad norm. Furthermore, while the addition of TSG does somewhat improve performance over just LN freeze and grad norm, such gains are relatively small and inconsistent across different conditions. This seems to be the case both when comparing TSG + LN freeze to LN freeze, TSG + grad norm to grad norm, and TSG + LN freeze + grad norm to LN freeze + grad norm.
3. As noted in the weaknesses, there is no causal link between TSG better approximating the effects of joint attention weight ablation and the increase in attribution faithfulness of GIM.

As it currently stands, despite the novelty of the self-repair mechanism for attention and the comprehensive benchmarks run across different models, my score is a 4. This is because the analysis into the novel discovery of the self-repair mechanism seems underexplored, the causal link between attention self-repair and the improvements in faithfulness via TSG are not established, and said improvements are relatively incremental upon prior methods.

**Questions:**

1. See Weaknesses.
2. For faithfulness per layer, do you have any data for the larger models such as Llama 3.1 8B or Gemma 2 9B? Even without the Integrated Gradients approach, which might take too long?

---

> ### Author Response · Authors · 2025-11-17
> **Revisions and response to weaknesses**
>
> Dear reviewer,
>
> Thank you for your thoughtful and detailed review of our paper. We appreciate your recognition of our work's strengths, particularly the mathematical clarity of the self-repair mechanism, the theoretical motivation, and GIM's comprehensive performance across multiple models and datasets.
>
> We have revised the paper based on your feedback and that of the other reviewer.
> Below, we address each of your concerns:
>
> 1. We acknowledge that deeper analysis would strengthen the paper. We conducted several additional experiments but ultimately excluded them due to inconclusive results:
>    1. We attempted to evaluate TSG on examples where attention self-repair occurs versus where it does not. However, self-repair occurs in every example, making this comparison infeasible.
>    2. We attempted to apply TSG only to attention scores where we detected self-repair during backpropagation; however, this worsened performance. The issue is that self-repair is continuous rather than binary. When using a threshold to detect self-repair, we amplify specific attention scores more than others, creating imbalances that result in some components receiving disproportionate attribution. This imbalance propagates through backpropagation to earlier layers.
>
> We welcome suggestions for additional experiments to analyze self-repair in attribution.
>
> 2. We have updated the ablation study in the revised paper. There are now three ablation study tables in the appendix (Tables 4, 5, and 6). In Table 4, we show the impact of ablating a modification from GIM. We show the mean values and the standard deviations. The ablations that deteriorate the performance of GIM are shown in bold. Those that are significant are underlined (independent t-test $0.05<\alpha$). Across most model-data pairs, all three modifications are important. For some model-data pairs, ablating a modification will improve the performance. Variance is expected, and in Table 5, we demonstrate this. Integrated gradients are generally considered superior to GradientXInput in the field. In Table 5, we compare the two methods. While integrated gradients are generally better, the results vary across models and datasets. This shows that we cannot expect improvements to be consistent across datasets and models. In Table 6 (what used to be Table 3), you are correct that TSG only improves performance when combined with Layernorm freeze and Grad norm. However, note that both Layernorm freeze and Grad norm are essential. Individually, they do not improve the results (there is a lot of variance, but in the big picture, they do not enhance performance). Layernorm freeze and Grad norm seem to stabilize backpropagation, allowing TSG to also improve performance. The modifications are beneficial together. We have emphasized this observation in the Appendix (A.2)
> 3. We agree that the lack of a causal link is a weakness. However, we don't believe this is sufficient grounds for rejection given our other contributions. Even if future work finds no causal link, we have discovered self-repair in attention, showed that it is frequent and causes zero-gradients in the attention scores. Furthermore, GIM, the combination of our three modifications, outperforms other circuit discovery methods and feature attribution methods.
>
> **Answer to question 2:**
>
> We are computing layer-wise AOPC for Gemma-9B and LLAMA-8B. We will update the paper as soon as we have the results. We are not computing integrated gradients because it is too slow to be computed before the discussion deadline.
>
> **Additional changes.**
>
> GIM substantially outperforms other circuit discovery methods on the mechanistic interpretability benchmark (A.1).  GIM being state-of-the-art on this benchmark is a significant contribution, but we added it to the appendix (A.1) rather than including it in the main paper. With the extra page in the camera-ready version, we will move these results from the appendix to the main paper and list them as one of our main contributions.

---

### Official Review · Reviewer_8bWm · 2025-10-31

**Soundness:** 3
**Presentation:** 3
**Contribution:** 3
**Rating:** 6
**Confidence:** 3

**Summary:**

The paper argues that standard gradient-based explanations can miss important attention behaviours in transformers due to “attention self-repair,” where multiple high-weight tokens with similar value contributions can swap probability mass without changing the output, driving per-score gradients toward zero. To address this, the authors propose a set of tweaks, raising the softmax temperature only during backpropagation to better capture joint effects among interacting scores, along with stabilising adjustments around LayerNorm and gradient scaling. Empirically, they report improved faithfulness metrics across several models and tasks, suggesting these modifications yield explanations that align better with the model’s true reliance on attention

**Strengths:**

The approach addresses redundancy-driven cancellation inside the attention mechanism, and integration seems lightweight e.g no model retraining and only small code edits.

The paper spans multiple model families and datasets with cumulative analyses. The Improvements in comprehensiveness/sufficiency suggest better alignment with model behaviour.

**Weaknesses:**

Most evaluations are short-context benchmarks; it’s unclear how well the approach holds up in longer contexts.

Section 5.2 introduces a 0.1 threshold to decide when to treat effects as joint, which helps but feels coarse, as it is a global heuristic with no principled grounding. Over and underaggregation might occur.

The approach should be tested to determine whether attributions stay stable under light, meaning-preserving edits (paraphrases, synonym swaps, punctuation/tokenisation noise) and simple adversarial tweaks. Otherwise, the reported gains may be brittle in realistic settings.

**Questions:**

N/A

---

> ### Author Response · Authors · 2025-11-17
>
> Thank you for your thoughtful review. We have revised the paper based on your feedback and that of the other reviewer. Below, we address the weaknesses:
>
> 1. Many of the datasets have quite long contexts. On average, BoolQ comprises 1184 tokens, Movie Review 930, SciFact 448, and Fever 336 (see Table 7). These datasets have longer contexts than what other explainability papers usually evaluate on [1-9]. Therefore, calling the lack of long-context evaluation a weakness seems unfair. That said, a paper that analyzes the effect of varying the length of the context would be interesting in future work.
>
> 2. You are correct that our threshold of 0.1 is not based on principled grounding. We found the threshold empirically by testing lower and higher thresholds. We have added this to the appendix (A.12). In Figure 18, we present the results obtained using thresholds of 0.05, 0.1, and 0.2. Using different thresholds does not significantly change the results. A higher threshold adds points to the figure, where the additional points have less self-repair.
> Note also that this threshold is for illustrative purposes. Self-attention is a continuous phenomenon rather than a discrete one, and our method does not depend on a threshold.
>
> 3. You write that if we do not evaluate on adversarial examples, we may risk brittleness in realistic settings. We disagree that not evaluating on adversarial examples means we have not evaluated the methods in realistic settings. Many of our datasets originate from realistic settings: HateExplain comprises racist, hateful, and poorly spelled tweets, while Twitter comprises tweets with numerous spelling errors. Furthermore, we evaluate the methods on the Mechanistic Interpretability Benchmark (A.1), which comprises multiple datasets and tasks. Our paper evaluates on far more tasks, datasets, and models than what most other explainability papers do [1-9]. The breadth of our experiments is a strength, not a weakness.
>
> **Additional changes.**
>
> GIM substantially outperforms other circuit discovery methods on the mechanistic interpretability benchmark (A.1).  GIM being state-of-the-art on this benchmark is a significant contribution, but we added it to the appendix (A.1) rather than including it in the main paper. With the extra page in the camera-ready version, we will move these results from the appendix to the main paper and list them as one of our main contributions.
>
>
> **References**
>
> [1] Mukund Sundararajan, Ankur Taly, and Qiqi Yan. Axiomatic Attribution for Deep Networks. In Proceedings of the 34th International Conference on Machine Learning, pp. 3319–3328. PMLR, July 2017.
>
> [2] Aaquib Syed, Can Rager, and Arthur Conmy. Attribution Patching Outperforms Automated Circuit Discovery,  In Proceedings of the 7th BlackboxNLP Workshop: Analyzing and Interpreting Neural Networks for NLP, pages 407–416, Miami, Florida, US. Association for Computational Linguistics. November 2024.
>
> [3] Michael Tsang, Sirisha Rambhatla, and Yan Liu. How does This Interaction Affect Me? Interpretable Attribution for Feature Interactions. In Advances in Neural Information Processing Systems, volume 33, pp. 6147–6159. Curran Associates, Inc., 2020.
>
> [4] Kevin Wang, Alexandre Variengien, Arthur Conmy, Buck Shlegeris, and Jacob Steinhardt. Interpretability in the Wild: A Circuit for Indirect Object Identification in GPT-2 small, The Eleventh International Conference on Learning Representations. 2023.
>
> [5] Cody Rushing and Neel Nanda. Explorations of Self-Repair in Language Models, ICML'24: Proceedings of the 41st International Conference on Machine Learning. May 2024.
>
> [6] Avanti Shrikumar, Peyton Greenside, and Anshul Kundaje. Learning important features through propagating activation differences. In Proceedings of the 34th International Conference on Machine Learning - Volume 70, ICML’17, pp. 3145–3153, Sydney, NSW, Australia, August 2017a. JMLR.org. Avanti Shrikumar, Peyton Greenside, Anna Shcherbina, and Anshul Kundaje. Not Just a Black Box: Learning Important Features Through Propagating Activation Differences, April 2017b.
>
> [7] Thomas McGrath, Matthew Rahtz, Janos Kramar, Vladimir Mikulik, and Shane Legg. The Hydra Effect: Emergent Self-repair in Language Model Computations, Arxiv. July 2023.
>
> [8] Kevin Meng, David Bau, Alex Andonian, and Yonatan Belinkov. Locating and Editing Factual Associations in GPT. Advances in Neural Information Processing Systems 35. 2022.
>
> [9] Qing Lyu, Marianna Apidianaki, and Chris Callison-Burch. Towards Faithful Model Explanation in NLP: A Survey. Computational Linguistics, 50(2):657–723, June 2024. doi: 10.1162/coli_a_00511.

---

### Meta-Review · Area_Chair_c2SY · 2025-12-04

**Summary:**

This paper took me quite some time to go through, and I spent considerable effort on it. I reached the conclusion of rejecting the paper as it currently stands.

First, I commend the authors on the great effort they have put into the paper and the rebuttal. The paper initially received mixed reviews; however, after the rebuttal, at least one reviewer seemed to have been convinced by the results and was willing to increase their score.

However, I still have some concerns. First, the core fix (TSG) is heuristic. Some evaluation choices (gradient-space diagnostics, fixed hyperparameters) raise robustness questions, and a few methodological choices (layernorm freeze, grad-norm) are ad hoc enough that they require extra validation. With the additional causal/forward checks and robustness sweeps suggested above, the paper would become much stronger and more persuasive.

Moreover, I think the authors could have provided a theoretical justification by looking at the weights of a single or two attention layers in which the network is equivalent by design to an OR function. Then, they would likely have been able to theoretically justify, without any hand-wavy approximation, their TSG or at least the temperature-scaling part of it.

Also, while the motivation for the problem is good, it makes assumptions whose practical prevalence is unclear. In Figure 1, attention self-repair is only a problem when two conditions are met in the forward pass: the value vectors are similar, and we are examining a single attention weight. In the example of Figure 1, if the values were set to [1, 2, 0], the overall weighted output would have changed.

As for the multiple layers, while the output in Figure 1 is indeed 1 despite the two different ablations, these are the representations at two different positions. Thus, when composed with further layers toward the final output, their impact on the output logits is far less obvious. This, of course, is somewhat captured by Equations (3) and (4), where in (4) you capture the idea that when the values are the same (they do not have to be identical, but this is one trivial way to satisfy the condition), the score depends on the gradient of the final logit z with respect to o.

Overall, I believe the paper contains promising ideas that could become impactful with additional theoretical grounding and more rigorous evaluation. I encourage the authors to address these issues and resubmit, as a strengthened version of this work would be a valuable contribution to the interpretability literature.

**Reviewer Concerns:**

see above

**Reviewer Scores:**

see above

---

### Decision · Program_Chairs · 2026-01-26

Reject